# What are Key Factors for Updates in RL for LLM Reasoning?

## Abstract

Reinforcement Learning from Verifiable Rewards (RLVR) has emerged as a promising framework for enhancing the reasoning ability of large language models. However, much of the existing work is guided by heuristic intuition, leading to divergent algorithmic choices, even contradictory ones that nevertheless report empirical gains. To better understand this phenomenon, we conduct a theoretical analysis of RLVR updates. Our study reveals that differences in off-policy degree, determined by the number of gradient steps per rollout, substantially affect the distribution of importance sampling ratios and their clipping behavior, thereby altering which tokens dominate the update. Building on this insight, we characterize gradient expectation as the central quantity governing update dynamics and analyze the roles of token probability, advantage, and importance sampling ratio. Motivated by these findings, we propose Adaptive Clip Policy Optimization (ACPO), which adjusts clipping boundaries across token groups according to the empirical variance of their importance sampling ratios. Experiments on models of varying scales (3B, and 7B) and diverse reasoning benchmarks—including mathematical problem solving, tabular question answering, and logic puzzles—show that ACPO outperforms strong baselines such as DAPO and CISPO. These results demonstrate that principled, analysis-driven approaches yield more robust and effective RLVR methods.

## 1 Introduction

Reinforcement Learning from Verifiable Rewards (RLVR) aims to enhance the reasoning ability of large language models (LLMs) by optimizing against checkable signals, such as mathematical correctness or logical validity (Shao et al., 2024; Guo et al., 2025a; Lambert et al., 2024; Jaech et al., 2024; Chen et al., 2025c; Yang et al., 2025a). This setting provides a scalable way to directly reward structured reasoning, making RLVR a central approach for advancing reliability in complex problem-solving tasks.

Despite lots of progress in this direction, much of the current work on RLVR remains driven by heuristic intuition rather than systematic analysis. As a result, a wide variety of algorithmic choices have been proposed (Yu et al., 2025; Yue et al., 2025; Su et al., 2025; Zheng et al., 2025a), but the effectiveness of these improvements is not well understood. There are even studies that adopt opposite approaches yet report empirical improvements. A notable example concerns how tokens with different probability or entropy levels are treated during optimization. Wang et al. (2025b) argues for emphasizing high-entropy tokens while masking low-entropy ones, on the premise that uncertain tokens are more important in the reasoning process. In contrast, Yang et al. (2025b) suggests emphasizing high-probability tokens to prevent low-probability tokens dominating updating, since low-probability token leads to larger gradients. Given the positive correlation between token entropy and probability, these strategies are effectively opposite in design, yet both have demonstrated benefits in practice.

Considering the importance of understanding RLVR updates at the token level (Yang et al., 2025b; Wang et al., 2025b; Cui et al., 2025; Chen et al., 2025a; Wang et al., 2025a), in this work, we conduct a detailed analysis of the update step in RLVR. Our investigation begins with a key observation: the two aforementioned studies have a key difference in their experimental settings. The work emphasizing high-entropy tokens performs 16 gradient steps per rollout step, resulting in an updating that

is closer to off-policy learning. In contrast, the work emphasizing high-probability tokens applies only 2 gradient steps per rollout step, which corresponds to a more on-policy update. This difference has a significant effect on the distribution of importance sampling (IS) ratios and the extent to which they are clipped. Since clipped tokens no longer contribute to the update, the critical insight is that *the analysis must move beyond the effect of individual tokens and instead consider the collective contribution of all tokens that actively participate in gradient updates, that is, the expectation of the gradient.* Building on this perspective, we provide a deeper theoretical analysis and demonstrate that greater degree of off-policy increases the variance of IS ratio, which in turn alters the gradient expectation and leads to different sets of tokens dominating the update under varying degrees of off-policy.

Extending the perspective of analyzing gradient expectation, we further analyze other key factors that influence the update dynamics in RLVR. These include (1) intrinsic token properties such as probability, (2) the sign and magnitude of the corresponding advantage, and (3) the importance sampling ratio. Taken together, these factors indicate that an effective update must account for their joint impact. In particular, the commonly used uniform clipping strategy fails to adequately reflect these heterogeneous influences. To address this limitation, we further propose Adaptive Clip Policy Optimization (**ACPO**). Rather than applying a fixed clipping range, ACPO calibrates the clipping boundary for different groups of tokens by setting it according to the empirical standard deviation of the importance sampling ratios within each group. This design is theoretically supported by our characterization of gradient expectation and provides a more principled mechanism for balancing stability and learning efficiency.

We validate the effectiveness of ACPO across models of different scales (3B, and 7B parameters) and a diverse set of reasoning tasks, including mathematical problem solving, tabular question answering, and logic puzzles. Experimental results demonstrate that ACPO consistently outperforms strong baselines such as DAPO (Yu et al., 2025) and CISPO (Chen et al., 2025a). These extensive evaluations highlight that a more comprehensive approach, derived from careful analysis, yields greater robustness and adaptability, enabling effective learning across varying base models and datasets.

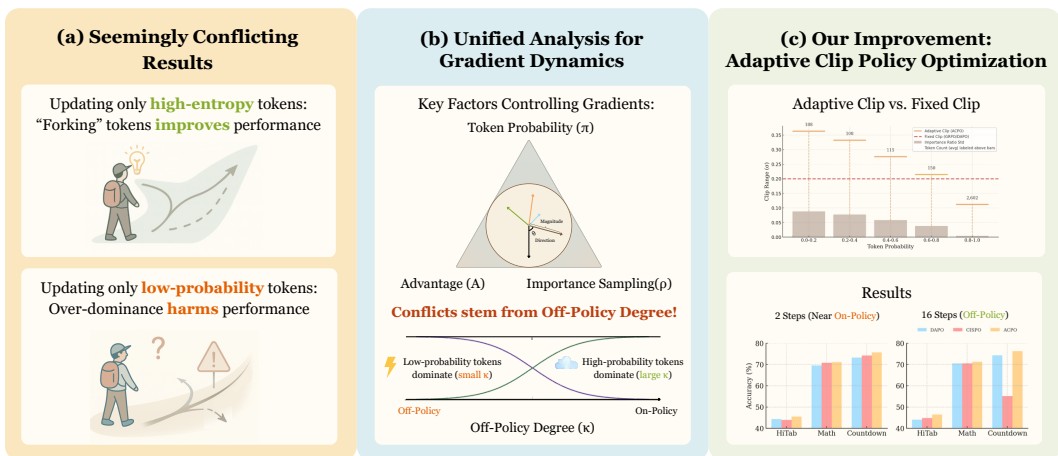

Figure 1: (a) Prior work shows seemingly conflicting results: updating high-entropy tokens improves performance, while over-dominance of low-probability tokens harms stability. (b) Our unified analysis reveals that these conflicts arise from gradient dynamics controlled by token probability, advantage, and importance sampling, shifting with the degree of off-policy divergence. (c) Based on these insights, we propose Adaptive Clip Policy Optimization (ACPO), which dynamically calibrates clipping ranges and outperforms fixed-clip baselines in both near on-policy and off-policy settings.

## 2 THEORETICAL ANALYSIS OF EMPIRICAL CONTRADICTIONS

### 2.1 PRELIMINARY

In contrast to traditional actor-critic algorithms such as Proximal Policy Optimization (PPO) (Schulman et al., 2017), Group Relative Policy Optimization (GRPO) (Shao et al., 2024) removes the critic

(of comparable size to the policy) and estimates advantages by standardizing rewards within a sampled group. For each prompt $q \sim P(Q)$, the policy $\pi_\theta$ samples $G$ responses $\{o_i\}_{i=1}^G$, each scored by a rule-based reward $r_i = R(o_i)$. The (intra-group) relative advantage is $A_i = \frac{r_i - \text{mean}(\{r_j\}_{j=1}^G)}{\text{std}(\{r_j\}_{j=1}^G)}$, where $\text{mean}$ and $\text{std}$ denote the sample mean and standard deviation, respectively. The objective is:

$$J_{\text{GRPO}}(\theta) = \mathbb{E}_{q \sim P(Q), \{o_i\}_{i=1}^G \sim \pi_{\theta_{\text{old}}}(\cdot|q)} \frac{1}{G} \sum_{i=1}^G \frac{1}{|o_i|} \sum_{t=1}^{|o_i|} \left[ \min(\rho_{i,t} A_i, \text{clip}(\rho_{i,t}, 1 - \epsilon_l, 1 + \epsilon_h) A_i) \right] \quad (1)$$

where $\rho_{i,t} = \frac{\pi_\theta(o_{i,t}|q,o_{i,<t})}{\pi_{\theta_{old}}(o_{i,t}|q,o_{i,<t})}$ is the importance sampling (IS) ratio, used to correct for off-policy updates. The clipping function $clip(\cdot, 1 - \epsilon_l, 1 + \epsilon_h)$ restricts this ratio to $[1 - \epsilon_l, 1 + \epsilon_h]$ for stable updates. GRPO typically uses symmetric clipping ($\epsilon_l = \epsilon_h$), while DAPO (Yu et al., 2025) adopts asymmetric clipping ($\epsilon_h > \epsilon_l$) to mitigate entropy collapse. Following recent practice (Yu et al., 2025; Su et al., 2025; Chen et al., 2025a), we omit the KL penalty.

## 2.2 CONTRADICTORY EMPIRICAL RESULTS

The limitations of a heuristic-driven approach are highlighted by recent, conflicting findings. Wang et al. (2025b) report faster convergence and superior accuracy by selectively updating high-entropy tokens, since these tokens tend to have higher uncertainty and play a more important role in the reasoning process. In contrast, Yang et al. (2025b) argue against amplifying the influence of low-probability tokens, demonstrating that exclusively updating them leads to slow convergence and suboptimal performance. Their work suggests that the large, high-variance gradients from these tokens can destabilize training. This creates a paradox, as high-entropy tokens are often those with low generation probabilities Guo et al. (2025b); Zheng et al. (2025b). The fact that two strategies targeting overlapping token populations yield divergent outcomes suggests that a critical factor is being overlooked. This puzzling discrepancy motivates our investigation into the fundamental structure of the policy gradient.

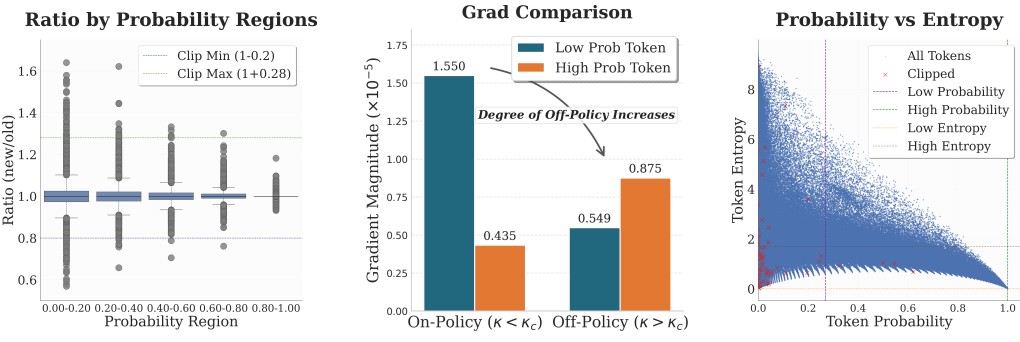

Figure 2: Overview of empirical validations for importance sampling ratio variance and gradient dominance. (a) Distribution of the importance sampling ratio, showing significantly higher variance for low-probability tokens. (b) The reversal of gradient dominance from low- to high-probability tokens as the off-policy degree $\kappa$ increases. (c) Correlation between token probability and entropy.

## 2.3 EXPLAIN THE DIFFERENCE VIA THEORETICAL ANALYSIS

### 2.3.1 FINDING: DEGREE OF OFF-POLICY IS THE KEY

We observe a substantial difference in the experimental setting of two closely related studies. In one (Yang et al., 2025b), a single rollout batch is used to perform two update steps; in the other (Wang et al., 2025b), the same batch is used for 16 updates. After the first update, the current policy diverges from the behavior policy that generated the rollouts, and additional updates enlarge

this divergence. Consequently, the two settings differ markedly in their degree of off-policy divergence. Moreover, as indicated by Eq. equation 1, the importance sampling (IS) ratio used to correct off-policy data is clipped, and tokens whose ratios are clipped will not contribute to the gradient update. Under different levels of off-policy divergence, therefore, both the number and the distribution of tokens that actually influence learning change, potentially leading to qualitatively different outcomes. These observations suggest that *analyses based solely on gradients from individual tokens are incomplete; a more comprehensive perspective is to examine the **gradient expectation** by aggregating contributions across all tokens that participate in the update.*

### 2.3.2 DETAILED ANALYSIS

The gradient of the GRPO objective with respect to the model parameters $\theta$ is given by:

$$\nabla_\theta J_{\text{GRPO}}(\theta) = \mathbb{E}_{q \sim P(Q), \{o_i\}_{i=1}^G \sim \pi_{\theta_{old}}(O|q)} \frac{1}{G} \sum_{i=1}^G \frac{1}{|o_i|} \sum_{t=1}^{|o_i|} \left[ \rho_{i,t} A_i \cdot \mathbb{I}_{\text{clip}}(\rho_{i,t}, A_i) \right] \cdot \nabla_\theta \log \pi_\theta(o_{i,t})$$

$$\text{where } \mathbb{I}_{\text{clip}}\left(\rho_{i,t}, \hat{A}_{i,t}\right) = \begin{cases} 0 & \begin{cases} \text{if } \hat{A}_{i,t} > 0 \text{ and } \rho_{i,t} > 1 + \epsilon_h \\ \text{if } \hat{A}_{i,t} < 0 \text{ and } \rho_{i,t} < 1 - \epsilon_l \end{cases} \\ 1 & \text{otherwise} \end{cases} . \tag{2}$$

To analyze the update on the model parameters $\theta$, our work focuses on the gradient with respect to the pre-softmax logits $z_k$. Via the chain rule, the full gradient $\nabla_\theta \log \pi_\theta(o_{i,t})$ in Eq. 2 is propagated from these logits. We analyze this initial update signal, specifically $\partial \log \pi_k / \partial z_k = (1 - \pi_k)$ for a sampled token $k$. In this setting, the indicator function $\mathbb{I}_{\text{clip}}$ set to zero means that the tokens is clipped and they have no contributions to the gradient. Therefore, the core factors that influence the gradient signal can be defined as:

$$G_k = (1 - \pi_k) \cdot \rho_k \cdot A \cdot \mathbb{I}_{\text{clip}}(r_k, A) \tag{3}$$

Our objective is to analyze the expectation of this quantity, $\mathbb{E}[G_k]$, particularly how it is influenced by the token probability $\pi_k$ and the degree of off-policy during training.

**Distribution of Importance Sampling Ratio**  To analyze the impact of the clipping mechanism on the gradient expectation $\mathbb{E}[G_k]$, it is important to characterize the distribution of the importance sampling ratio, $\rho_k$. Assume $\rho_k$ follows a normal distribution, $\mathcal{N}(1, \sigma_\rho^2)$. Its mean is 1 as the policy update step can stochastically increase or decrease a token's probability. Then, the primary challenge lies in determining its variance, $\sigma_\rho^2$, which dictates the likelihood of clipping. By performing a second-order Taylor expansion of the policy function, we approximate how $\sigma_\rho^2$ evolves from a single gradient update step (see Appendix B.1 for details). This derivation yields our first key result:

**Lemma 1 (Variance of Importance Sampling Ratio).** *Consider a single-step gradient update from policy $\pi_{old}$ to $\pi$, the variance of the importance sampling ratio, $\sigma_\rho^2$, can be expressed as a function of the token probability $\pi_k$ as follows:*

$$\sigma_\rho^2(\pi_k) = \kappa^2(1 - \pi_k)^4 + O(\kappa^3) \tag{4}$$

*Here, $\kappa = (\frac{\eta}{T^2})\sigma_A$ is a coefficient related to the learning rate $\eta$, the temperature $T$ and the advantage variance $\sigma_A^2$. A larger $\kappa$ corresponds to a greater per-step off-policy shift, since for the same token probability $\pi_k$, increasing $\kappa$ proportionally enlarges $\sigma_\rho^2(\pi_k)$. The term $O(\kappa^3)$ represents higher-order terms, which are negligible under the standard assumption that the learning step is small (i.e., $\kappa \ll 1$).*

Figure 2 provides an empirical visualization of this principle. The distribution of the IS ratio for low-probability tokens is shown to have a much larger spread and numerous outliers compared to that of high-probability tokens.

**Expectation of Gradient**  With the distribution of the IS ratio established in Lemma 1, we can now derive an analytical approximation for the gradient expectation magnitude, $\mathbb{E}[G_k]$. We assume the advantage $A$ follows a zero-mean normal distribution, $\mathcal{N}(0, \sigma_A^2)$. The full derivation is provided in Appendix B.2.

**Proposition 1 (Gradient Expectation Magnitude).** *For a token with probability $\pi$, its gradient Expectation magnitude $\mathbb{E}[G|\pi]$ can be expressed as:*

$$\mathbb{E}[G \mid \pi] = (1 - \pi) \frac{\sigma_A}{\sqrt{2\pi}} \cdot F(\pi; \kappa, \epsilon_h, \epsilon_l) \tag{5}$$

*where the function $F(\cdot)$ is defined as:*

$$F(\pi; \kappa, \epsilon_h, \epsilon_l) = \Phi\left(\frac{\epsilon_h}{\sigma_\rho(\pi)}\right) - \Phi\left(\frac{\epsilon_l}{\sigma_\rho(\pi)}\right) - \sigma_\rho(\pi)\left[\phi\left(\frac{\epsilon_h}{\sigma_\rho(\pi)}\right) + \phi\left(\frac{\epsilon_l}{\sigma_\rho(\pi)}\right)\right] \tag{6}$$

*Here, $\Phi(\cdot)$ and $\phi(\cdot)$ are the CDF and PDF of the standard normal distribution, respectively. 1.*

The function $F(\cdot)$ captures the effect of clipping. Its behavior is governed by the standardized clipping thresholds, $\epsilon_h/\sigma_\rho(\pi)$ and $\epsilon_l/\sigma_\rho(\pi)$.

**Two Different Gradient Dominances** Proposition 1 provides the analytical tool to explain the empirical contradictions outlined in Section 2.2. We partition tokens into two groups based on their predicted probability $\pi$: low-probability tokens with $\pi \in [0, p_L]$ and high-probability tokens with $\pi \in [p_H, 1)$, for some thresholds $p_L < p_H$. We then compare the average gradient expectation magnitudes of low-probability tokens, denoted $|\bar{G}_L|$, with those of high-probability tokens, $|\bar{G}_H|$ as follows (detailed derivations are in Appendix B.3).

**Remark 1 (Gradient Dominance Reversal).** *There exists a critical threshold for the degree of off-policy, $\kappa_c > 0$, that dictates a reversal in which token population dominates the gradient expectation:*

1. ***Near On-Policy** ($\kappa < \kappa_c$): The average gradient expectation from low-probability tokens are greater in magnitude than from high-probability tokens, i.e., $|\bar{G}_L| > |\bar{G}_H|$. Low-probability tokens dominate the updates.*

2. ***Sufficiently Off-Policy** ($\kappa > \kappa_c$): The relationship inverts, with $|\bar{G}_H| > |\bar{G}_L|$. High-probability tokens now dominate.*

This corollary offers a theoretically-grounded resolution to the empirical conflict, and the predicted dominance reversal is empirically validated in Figure 2. We can now contextualize the conflicting findings of prior work. The training regime of Yang et al. (2025b), with its small number of updates per batch, operates in the **near on-policy** region (small $\kappa$), where low-probability tokens provide the dominant gradient signal according to our analysis. Conversely, the method from Wang et al. (2025b) employs a high number of updates, pushing the training into a **sufficiently off-policy** regime (large $\kappa$). In this scenario, the extreme IS ratio variance of low-probability tokens leads to frequent clipping that suppresses their gradient contribution and high-probability dominate updating. Furthermore, the choice of entropy as a heuristic is particularly effective because, as shown in Figure 2 right, it also implicitly filters out the most unstable low-probability, low-entropy tokens (see Appendix C for a detailed analysis). Therefore, targeting high-entropy tokens balances contributions from different token populations, allowing for more stable updates.

Our analysis reveals that the question of which tokens are more important for optimization does not have a static answer. Instead, the optimal focus of gradient updates is conditional on the degree of off-policy during training.

## 3 KEY FACTORS INFLUENCING UPDATES IN RL FOR LLMS

Following the perspective of analyzing gradients and learning dynamics of GRPO, we further provide a more comprehensive investigation on the key factors governing the GRPO policy gradient.

### 3.1 ANALYTICAL FRAMEWORK

We analyze the gradients in two aspects: its magnitude and direction. First, we measure gradient magnitude via the gradient expectation, $\mathbb{E}[G|\pi]$ (Proposition 1), to identify which token populations dominate the update. Second, to analyze the high-dimensional gradient direction, we adopt

a layer-wise geometric approach. We use Singular Value Decomposition (SVD) to identify dominant update directions within a layer's gradient matrix and quantify their alignment using Principal Angles Between Subspaces (PABS).

For a gradient matrix $W \in \mathbb{R}^{m \times n}$, its SVD is $W = U \Sigma V^\top$. We focus on the subspace spanned by the top-$k$ right singular vectors, $V_k$. Given two such subspaces from different gradients, represented by orthonormal bases $Q^A$ and $Q^B$, the $k$ principal angles are:

$$\theta_i = \arccos(s_i), \quad \text{for } i = 1, \ldots, k, \tag{7}$$

where $s_i$ are the singular values of the cross-Gram matrix $(Q^A)^\top Q^B$. An angle of $0°$ indicates perfect alignment, while $90°$ implies orthogonality. We set $k = 128$ and clamp singular values to $[0, 1]$ for numerical stability.

## 3.2 KEY FACTORS

The GRPO gradient (Eq. 2) is shaped by three principal components, each influencing its final magnitude and direction. First, the **token's properties** determine the base update vector $\nabla_\theta \log \pi_\theta(o_{i,t})$. Second, the **advantage** $A_i$ acts as a weight. Its magnitude scales the update, and its sign determines whether the action is reinforced or suppressed. Finally, the **importance sampling** ratio $\rho_{i,t}$ and its associated clipping mechanism further modulate the update, selectively filtering contributions to correct for the off-policy mismatch and fundamentally altering the final gradient's composition. In the following sections, we dissect each of these components.

**Token Properties** The policy gradient is primarily driven by signals from a small subset of low-probability tokens. Our analysis (Appendix D.1) shows these tokens contribute a gradient with disproportionately large magnitude and strong alignment with the overall update direction, whereas high-probability tokens have a much smaller contribution. This explains why heuristics like updating high-entropy tokens are effective: they implicitly balance the potent but potentially unstable signals from the distribution's tail against policy stability. This underscores the need for a more principled mechanism to manage this trade-off.

**Sign of Advantage** The advantage sign introduces a critical asymmetry in the update dynamics. While positive and negative advantage samples yield gradients of comparable magnitude, by analyzing the direction, we find that the negative-advantage gradient is substantially better aligned with the final update direction (see Appendix D.2 for details). This indicates that the learning trajectory might be predominantly shaped by corrective signals from suboptimal actions ($A < 0$) rather than by reinforcing optimal ones ($A > 0$). This finding highlights the importance of negative samples in RL, which is a key differentiation with supervised learning, where the model only learns to fit positive samples.

**Importance Sampling** The clipping mechanism in off-policy learning, while essential for stability, fundamentally alters the optimization path. By comparing the direction between the off-policy gradient and on-policy gradient by aggregating all samples in one step, we find that they diverge significantly ($\approx 47°$) even when their magnitudes are nearly identical (see Appendix D.3 for more details). This demonstrates that clipping is not a neutral stabilizer; it actively shapes the gradient by systematically filtering high-variance updates, prioritizing magnitude stability at the cost of directional fidelity to the on-policy ideal. This trade-off, further highlighted by comparing different clipping strategies, suggests an opportunity for adaptive mechanisms that can better preserve on-policy direction while retaining off-policy efficiency.

## 3.3 SUMMARY OF ANALYSIS

Our analysis of the policy gradient dynamics yields the following key insights. *(1) Uniformly clipping for all tokens is suboptimal*, since it ignores the probability-dependent nature of importance sampling variance (Lemma 1), treating all tokens uniformly despite their vastly different contributions to gradient instability. *(2) Clipping stabilizes updating but alters optimization direction.* Although clipping is critical for variance control, it can largely change gradient direction compared with pure on-policy updates. This reflects an inherent trade-off: enforcing stability through clipping reshapes the optimization trajectory, potentially sacrificing alignment with the on-policy update, also motivates us to introduce more careful designs on clipping.

Besides clipping, we also have some observations on other factors. *(1) Low-probability tokens dominate the gradient.* The update signal is disproportionately driven by these tokens, which are the major information source of updating, but may cause unstable learning signal. *(2) Samples with negative advantage play a critical role in shaping the update direction.* The learning trajectory is primarily dictated by corrective signals from negative-advantage samples ($A < 0$), suggesting that error correction is more influential than simple reinforcement in defining the optimization path.

# 4 ADAPTIVE CLIP POLICY OPTIMIZATION

Building upon the theoretical analysis (Section 2.3), empirical observations (Figure 2), and analysis on gradients (Section 3), a key insight is that the variance of the importance sampling (IS) ratio is not static. Instead, it is dynamically influenced by multiple factors, including a token's intrinsic probability, the degree of off-policy, and the current training stage.

Despite this inherent dynamism, most existing methods employ a fixed clipping range. This uniform approach is suboptimal, as it fails to account for the heterogeneous stability profiles of different tokens under varying conditions. To address this limitation, we propose **Adaptive Clip Policy Optimization (ACPO)**, a method that dynamically calibrates the clipping boundaries in proportion to the empirically measured standard deviation of the IS ratio. ACPO starts from a more comprehensive consideration on token properties, and thus is designed to adapt to various learning process with different models and datasets.

## 4.1 METHOD

Inspired by our analysis in Lemma 1, which shows that IS ratio variance is highly dependent on token probability, ACPO replaces the fixed, global clipping range with one tailored for different token groups. Specifically, tokens within a batch are grouped into bins (five in experiments) based on their generation probabilities. The ACPO objective is then applied to each bin:

$$J_{ACPO}(\theta) = \mathbb{E}_{q \sim P(Q), \{o_i\}_{i=1}^{G} \sim \pi_{\theta_{old}}(O|q)} \frac{1}{G} \sum_{i=1}^{G} \frac{1}{|o_i|} \sum_{t=1}^{|o_i|} \{min[\rho_{i,t}A_i, clip(\rho_{i,t}, 1-\epsilon_a, 1+\epsilon_a)A_i]\} \quad (8)$$

The adaptive clipping range $\epsilon_a$ is computed dynamically for each bin as $\epsilon_a = \alpha \cdot \sigma_\rho + \epsilon_b$. Here, $\sigma_\rho$ is the empirical standard deviation of IS ratios within the bin, measuring the local policy shift. The hyperparameter $\alpha$ scales the sensitivity to this shift; we find $\alpha = 3$ to be effective, inspired by the three-sigma rule. The term $\epsilon_b$ is a small base value that ensures a minimum clipping range for stability, preventing the trust region from shrinking too much. This formulation allows the optimization to self-regulate: it automatically widens the trust region to accommodate larger updates when the policy shift is high and tightens it for stable fine-tuning as the policy converges.

## 4.2 EXPERIMENTAL SETUP AND RESULTS

We implement our method on three reasoning tasks: **ORZ-57K** (Hu et al., 2025) for mathematical problem solving, **HiTab** (Cheng et al., 2022) for tabular question answering, and **Countdown** (Jackson, 2025) for arithmetic-based puzzles. Experiments are conducted with `Qwen2.5-3B` (Yang et al., 2024), `Qwen2.5-3B-Instruct`, and `Qwen2.5-7B` to examine its efficiency and scalability. We compare our method against two strong baselines: **DAPO** (Yu et al., 2025), an adaptive variant derived from GRPO (since our focus is specifically on the effect of clip ranges, we only apply its clip-higher component while keeping the rest identical to vanilla GRPO), and **CISPO** (Chen et al., 2025a), which modifies REINFORCE by clipping only the importance sampling ratios rather than gating token updates, thereby stabilizing training. Following Yang et al. (2025b) and Wang et al. (2025b), we evaluate both **near On-Policy** (N-OnP, 2-step update per rollout) and sufficiently **Off-Policy** (OffP, 16-step update per rollout) regimes to assess whether our method adapts effectively under different levels of policy divergence. For models trained on **ORZ-57K**, we additionally evaluate their out-of-distribution generalization on other mathematical reasoning benchmarks, including **AMC2023**, **Minerva**, **Math500** (Hendrycks et al., 2021), and **OlympiadBench** (He et al., 2024), in order to verify robustness across diverse math domains.

Table 1: Results trained on Countdown and HiTab datasets. We report the best validation reward under N-OnP. and OffP. training regimes, with results given as the average accuracy over 4 evaluations per benchmark.

(a) Performance of Qwen2.5-3B-Instruct model trained on the Countdown dataset.

| Method | Qwen2.5-3B-Instruct | |
|---|---|---|
| | N-OnP. | OffP. |
| Base Model | 6.90 | |
| DAPO | 73.27 | 74.38 |
| CISPO | 74.25 | 55.12 |
| ACPO | **75.74** | **76.27** |

(b) Performance of Qwen2.5-3B and Qwen2.5-7B model trained on the HiTab dataset.

| Method | Qwen2.5-3B | | Qwen2.5-7B | |
|---|---|---|---|---|
| | N-OnP. | OffP. | N-OnP. | OffP. |
| Base Model | 17.50 | | 31.25 | |
| DAPO | 44.33 | 44.00 | 68.42 | 65.75 |
| CISPO | 43.92 | 44.83 | 69.17 | 65.67 |
| ACPO | **45.50** | **46.42** | **69.83** | **66.58** |

Table 2: Results of Qwen2.5-7B model trained on the ORZ-57K dataset, reported as average accuracy over 8 evaluation runs across benchmarks.

| Method | AMC2023 | | Minerva | | Math500 | | Olympiad | | Avg. |
|---|---|---|---|---|---|---|---|---|---|
| | N-OnP. | OffP. | N-OnP. | OffP. | N-OnP. | OffP. | N-OnP. | OffP. | |
| Base Model | 26.20 | | 13.32 | | 41.48 | | 18.02 | | 24.76 |
| DAPO | 41.11 | 42.47 | 25.74 | **26.19** | 69.55 | 70.53 | 31.63 | 31.94 | 42.40 |
| CISPO | 39.46 | **43.37** | 26.24 | 26.15 | 70.80 | 70.58 | 31.46 | 31.72 | 42.47 |
| ACPO (Ours) | **41.42** | 42.62 | **26.52** | 26.15 | **71.15** | **71.35** | **31.76** | **32.26** | **42.90** |

**Main Results**. **Tables 1–2** and **Figure 3** demonstrate that ACPO consistently outperforms baseline methods across all three reasoning tasks and model scales. The improvements are particularly notable in challenging off-policy scenarios: ACPO achieves the highest performance on ORZ (73.2%), HiTab (69.83%), and Countdown (76.27%).

A key observation is ACPO's stable performance across both near on-policy and off-policy training regimes, while baselines show varying degrees of instability. Most notably, CISPO suffers a dramatic performance collapse on Countdown (from 74.25% to 55.12%), illustrating the brittleness of heuristic-based approaches when policy divergence increases. In contrast, ACPO maintains consistent gains in both near on-policy and sufficiently off-policy settings.

These results validate our theoretical framework in two critical ways. First, the consistent cross-regime performance confirms that our adaptive clipping mechanism successfully handles the gradient dominance reversal predicted in Corollary 1. Second, the unified improvements across diverse reasoning tasks demonstrate that context-aware optimization provides more robust benefits than static, heuristic-based modifications. The experimental evidence thus supports our central claim that effective RLVR requires principled, adaptive approaches rather than fixed update strategies.

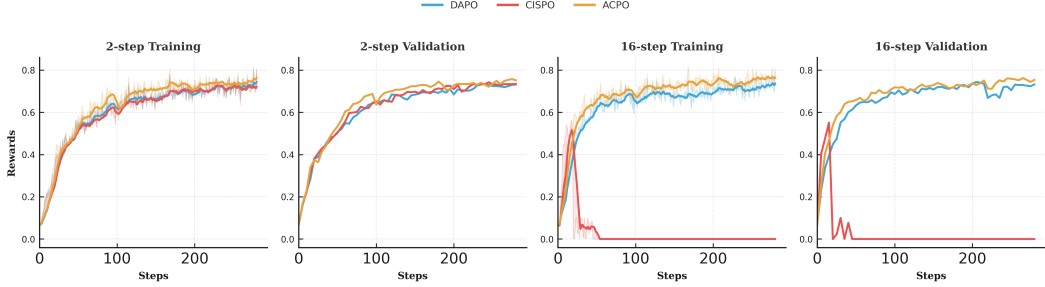

Figure 3: Training curves and validation rewards on Countdown dataset with near on-policy and off-policy updates.

Table 3: Ablation study results on the Countdown benchmark with Qwen2.5-3B-Instruct.

| Method / Setting | N-OnP. | OffP. |
|---|---|---|
| DAPO (Baseline) | 73.27 | 74.38 |
| DAPO (Clip-High=Max) | 74.65 | 75.86 |
| GRPO (Clip=Avg) | 72.55 | 74.16 |
| **ACPO** | **75.74** | **76.27** |

## 4.3 ABLATION STUDY

To isolate the contribution of adaptive clipping, we conduct ablation experiments on the Countdown task with Qwen2.5-3B-Instruct (Table 3). We evaluate two controlled settings for DAPO and GRPO: (1) **DAPO (Clip-High=Max)**, where the upper threshold is set to ACPO's maximum observed clip range, while keeping the lower bound unchanged. These settings ensure that any performance difference cannot be attributed to trivial changes in clipping magnitudes. (2) **GRPO (Clip=Avg)**, where GRPO's upper and lower clipping thresholds are both set to the weighted average of ACPO's per-token clip ranges.

Under the *avg* setting, ACPO's average clip range yields a uniform threshold that is **more restrictive** than standard GRPO, since it is dominated by high-probability tokens with low variance. This overly tight bound suppresses valuable updates from low-probability tokens, thus reducing performance (72.55% vs. 73.27% baseline).

Under the *max* setting, raising DAPO's clip-range upper bound to match ACPO's maximum observed range admits more signals and yields modest gains (74.65% vs. 73.27%). However, this uniform expansion ignores token-level variance: some tokens remain over-clipped, while others become overly permissive, leaving performance still below ACPO's dynamic calibration.

This gap demonstrates that ACPO's benefit stems from adaptive calibration rather than simply using different threshold values. Unlike static approaches that apply uniform bounds across all tokens, ACPO's variance-aware mechanism enables token-specific adjustments that balance stability with gradient informativeness, leading to consistent improvements across training regimes.

## 5 RELATED WORKS

**Reinforcement Learning for Large Language Models.** The application of reinforcement learning to LLMs has transitioned from preference alignment via RLHF (Ouyang et al., 2022) to directly optimizing reasoning abilities with Reinforcement Learning from Verifiable Rewards (RLVR) (Lambert et al., 2024). This paradigm uses objective feedback signals, such as mathematical correctness, to train models. Pioneered by early work like OpenAI's o1 (Jaech et al., 2024) and advanced by algorithms such as GRPO (Shao et al., 2024), RLVR has become a central methodology for developing state-of-the-art reasoning agents, demonstrating that complex skills can emerge from outcome-based optimization.

**Token-Level Update Strategies.** The success of RLVR is highly dependent on the token-level update mechanism. A diverse range of heuristic strategies has been proposed, focusing on areas like entropy-based modulation (Wang et al., 2025b; Cui et al., 2025), gradient clipping (Chen et al., 2025a; Zheng et al., 2025a), and advantage function design. However, these methods often yield conflicting conclusions. For instance, a key debate exists on whether to prioritize high-entropy tokens to encourage exploration in reasoning (Wang et al., 2025b) or to suppress them to mitigate high-variance gradients (Yang et al., 2025b). Such contradictions highlight a lack of a unified understanding of RLVR's update dynamics, motivating the need for the systematic, component-wise analysis presented in our work.

## 6 CONCLUSION

In this work, we moved beyond the heuristic-driven design of RLVR algorithms by conducting a systematic analysis of token-level update dynamics. Our core contribution is revealing that the degree of off-policy learning reconciles contradictory findings in prior work, such as the debate over

prioritizing high-entropy versus high-probability tokens. By analyzing the gradient expectation, we show that off-policy updates alter the distribution of importance sampling ratios, which in turn determines which tokens dominate the learning process. This insight exposes the limitations of uniform clipping and motivates our proposed method, Adaptive Clip Policy Optimization (ACPO), which calibrates clipping boundaries for different token groups based on their empirical IS ratio statistics. Extensive experiments across various models (3B/7B) and diverse reasoning tasks demonstrate that ACPO consistently outperforms strong baselines, underscoring that a principled, analytical approach is crucial for developing more robust and effective methods to enhance the reasoning capabilities of large language models.

## ETHICS STATEMENT

The primary objective of this work is to enhance the systematic understanding and reliability of reasoning in large language models. We view this as a crucial contribution to the development of safer and more predictable AI. All experiments were conducted on publicly available, standard academic datasets and utilized open-source base models. These resources do not contain any personally identifiable information or sensitive user data. While the base models may inherit societal biases from their pre-training data, our research focuses on optimizing for objective, verifiable tasks such as mathematical and logical correctness. This problem formulation is less susceptible to subjective biases, though we acknowledge that addressing underlying model bias remains a critical and ongoing challenge for the community. We also recognize that advancements in AI capabilities carry a potential for dual use, and we encourage the responsible development and deployment of these technologies. Our work is intended for a research audience to foster a deeper, more principled understanding of AI systems.

## REPRODUCIBILITY

To ensure the reproducibility of our findings, we provide comprehensive details of our training process, including all hyperparameters and experimental configurations, in Appendix F. Furthermore, our code and training scripts are made publicly available at the following repository: `https://anonymous.4open.science/r/ACPO`.

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

## A USE OF LARGE LANGUAGE MODELS (LLMs) STATEMENT

We used Large Language Models (LLMs) only as auxiliary tools during the preparation of this paper. Specifically, LLMs were employed for proofreading, grammar correction, and improving the readability of sentences. They were also occasionally used to assist with LaTeX formatting (e.g., tables and figures). All technical ideas, experimental design, analysis, and scientific claims are solely the work of the authors.

The authors take full responsibility for the content of the manuscript, including any text generated or polished by the LLM. We have ensured that the LLM-generated text adheres to ethical guidelines and does not contribute to plagiarism or scientific misconduct.

## B DETAILED THEORETICAL DERIVATIONS

### B.1 DERIVATION OF LEMMA 1

This appendix provides a detailed derivation for the relationship between the variance of the importance sampling ratio, $\sigma_\rho^2$, and the token probability, $\pi_k$, as stated in Lemma 1. We begin by performing a second-order Taylor expansion of the policy function to approximate its change after a single gradient update.

#### B.1.1 SECOND-ORDER TAYLOR EXPANSION OF THE POLICY

Let $\pi_k$ be the probability of sampling token $k$ from the vocabulary, which is produced by a softmax function with temperature $T$ over the logits $y$: $\pi_k = \frac{\exp(y_k/T)}{\sum_j \exp(y_j/T)}$. A single-step policy gradient update changes the logits $y_i$ by $\Delta y_i$. The change in the probability of token $k$, $\pi_k$, can be approximated by a second-order Taylor expansion with respect to the logit changes:

$$\pi'_k \approx \pi_k + \sum_i \frac{\partial \pi_k}{\partial y_i} \Delta y_i + \frac{1}{2} \sum_{i,j} \frac{\partial^2 \pi_k}{\partial y_i \partial y_j} \Delta y_i \Delta y_j \tag{9}$$

where $\pi'_k$ is the updated probability.

The gradient of the softmax function with temperature $T$ is:

$$\frac{\partial \pi_k}{\partial y_i} = \frac{1}{T} \pi_k (\delta_{ik} - \pi_i) \tag{10}$$

where $\delta_{ik}$ is the Kronecker delta.

The logit update $\Delta y_i$ from a single policy gradient step (e.g., in PPO) is given by:

$$\Delta y_i = \eta \cdot \frac{A}{T} \rho_k \cdot (\delta_{ik} - \pi_i) \tag{11}$$

where $\eta$ is the learning rate, $A$ is the estimated advantage for sampling token $k$, and $\rho_k := \pi'_k/\pi_k$ is the importance sampling ratio we aim to analyze.

**First-order term:** The first-order term of the Taylor expansion is:

$$\sum_i \frac{\partial \pi_k}{\partial y_i} \Delta y_i = \frac{\partial \pi_k}{\partial y_k} \Delta y_k + \sum_{i \neq k} \frac{\partial \pi_k}{\partial y_i} \Delta y_i \tag{12}$$

$$= \left( \frac{1}{T} \pi_k (1 - \pi_k) \right) \left( \eta \frac{A}{T} \rho_k (1 - \pi_k) \right) + \sum_{i \neq k} \left( -\frac{1}{T} \pi_k \pi_i \right) \left( \eta \frac{A}{T} \rho_k (-\pi_i) \right) \tag{13}$$

$$= \frac{\eta A \rho_k}{T^2} \pi_k (1 - \pi_k)^2 + \frac{\eta A \rho_k}{T^2} \pi_k \sum_{i \neq k} \pi_i^2 \tag{14}$$

$$\approx \frac{\eta A \rho_k}{T^2} \pi_k (1 - \pi_k)^2 \tag{15}$$

The approximation in the last step holds because $\sum_{i \neq k} \pi_i^2$ is typically small, especially when the vocabulary size is large and probabilities are diffuse. For simplicity and consistency with your draft, we will proceed with only the dominant term related to the update of the chosen token's logit, which is a common simplification.

**Second-order term:** For the second-order term, we focus on the diagonal element of the Hessian corresponding to the sampled token $k$ (i.e., $i = j = k$), as it contributes most significantly to the change. The second derivative is:

$$\frac{\partial^2 \pi_k}{\partial y_k^2} = \frac{\partial}{\partial y_k}\left(\frac{1}{T}\pi_k(1-\pi_k)\right) \tag{16}$$

$$= \frac{1}{T}(1-2\pi_k)\frac{\partial \pi_k}{\partial y_k} \tag{17}$$

$$= \frac{1}{T^2}\pi_k(1-2\pi_k)(1-\pi_k) \tag{18}$$

The contribution from this term (omitting the $1/2$ factor for simplicity in this approximation, following the draft) is:

$$\frac{\partial^2 \pi_k}{\partial y_k^2}(\Delta y_k)^2 = \left[\frac{1}{T^2}\pi_k(1-2\pi_k)(1-\pi_k)\right]\left[\eta\frac{A}{T}\rho_k(1-\pi_k)\right]^2 \tag{19}$$

$$= \frac{\eta^2 A^2 \rho_k^2}{T^4}\pi_k(1-2\pi_k)(1-\pi_k)^3 \tag{20}$$

Combining the first-order and second-order terms, the updated probability is:

$$\pi_k' \approx \pi_k + \frac{\eta A \rho_k}{T^2}\pi_k(1-\pi_k)^2 + \frac{\eta^2 A^2 \rho_k^2}{T^4}\pi_k(1-2\pi_k)(1-\pi_k)^3 \tag{21}$$

Dividing by $\pi_k$ gives an equation for the importance sampling ratio $\rho_k = \pi_k'/\pi_k$:

$$\rho_k \approx 1 + \left(\frac{\eta A}{T^2}\right)\rho_k(1-\pi_k)^2 + \left(\frac{\eta A}{T^2}\right)^2 \rho_k^2(1-2\pi_k)(1-\pi_k)^3 \tag{22}$$

To simplify, let's define the following variables:

- $K_1 = \frac{\eta A}{T^2}$
- $K_2 = (1-\pi_k)^2$
- $K_3 = (1-2\pi_k)(1-\pi_k)^3$

Substituting these into the equation for $\rho_k$ yields:

$$\rho_k \approx 1 + K_1 K_2 \rho_k + K_1^2 K_3 \rho_k^2 \tag{23}$$

Rearranging this gives a quadratic equation in $\rho_k$:

$$(K_1^2 K_3)\rho_k^2 + (K_1 K_2 - 1)\rho_k + 1 = 0 \tag{24}$$

### B.1.2 Variance of the Importance Sampling Ratio

We now estimate the variance of $\rho_k$, denoted as $\sigma_\rho^2$. We treat the advantage $A$ as a random variable with variance $\sigma_A^2$. Using the principle of **propagation of uncertainty**, the variance of $\rho_k$ can be approximated as:

$$\sigma_\rho^2 \approx \left(\frac{\partial \rho_k}{\partial A}\right)^2 \sigma_A^2 \tag{25}$$

We find the derivative $\frac{\partial \rho_k}{\partial A}$ using implicit differentiation on Equation 24. Differentiating both sides with respect to $A$:

$$\frac{\partial}{\partial A}\left[(K_1^2 K_3)\rho_k^2 + (K_1 K_2 - 1)\rho_k + 1\right] = 0 \tag{26}$$

Since $K_1$ is a function of $A$ ($\frac{\partial K_1}{\partial A} = \frac{\eta}{T^2}$), while $K_2$ and $K_3$ are not, applying the chain and product rules gives:

$$\left( 2K_1 \frac{\partial K_1}{\partial A} K_3 \rho_k^2 + K_1^2 K_3 \cdot 2\rho_k \frac{\partial \rho_k}{\partial A} \right) + \left( \frac{\partial K_1}{\partial A} K_2 \rho_k + (K_1 K_2 - 1) \frac{\partial \rho_k}{\partial A} \right) = 0 \qquad (27)$$

Grouping terms with $\frac{\partial \rho_k}{\partial A}$:

$$\frac{\partial \rho_k}{\partial A} (2\rho_k K_1^2 K_3 + K_1 K_2 - 1) = - \left( 2K_1 K_3 \rho_k^2 \frac{\partial K_1}{\partial A} + K_2 \rho_k \frac{\partial K_1}{\partial A} \right) \qquad (28)$$

Solving for $\frac{\partial \rho_k}{\partial A}$:

$$\frac{\partial \rho_k}{\partial A} = -\frac{\frac{\eta}{T^2}(2K_1 K_3 \rho_k^2 + K_2 \rho_k)}{2\rho_k K_1^2 K_3 + K_1 K_2 - 1} \qquad (29)$$

To simplify this expression, we consider the case of a small learning step, which is a standard assumption in such analyses. For a small update, the advantage $A$ is typically close to its mean (often assumed to be zero), and the policy does not change much, so $\rho_k \approx 1$. Under the approximation that $A \to 0$, we have $K_1 = \frac{\eta A}{T^2} \to 0$. The expression for the derivative simplifies significantly:

$$\lim_{A \to 0} \frac{\partial \rho_k}{\partial A} = -\frac{\frac{\eta}{T^2}(0 + K_2 \cdot 1)}{0 + 0 - 1} \qquad (30)$$

$$= \frac{\eta}{T^2} K_2 \qquad (31)$$

$$= \frac{\eta}{T^2} (1 - \pi_k)^2 \qquad (32)$$

Now, we can approximate the variance of the importance sampling ratio:

$$\sigma_\rho^2(\pi_k) \approx \left( \frac{\partial \rho_k}{\partial A} \right)^2 \sigma_A^2 \qquad (33)$$

$$\approx \left( \frac{\eta}{T^2} (1 - \pi_k)^2 \right)^2 \sigma_A^2 \qquad (34)$$

$$= \left( \frac{\eta}{T^2} \right)^2 \sigma_A^2 (1 - \pi_k)^4 \qquad (35)$$

By defining the coefficient $\kappa := \frac{\eta}{T^2} \sigma_A$, we arrive at our final result:

$$\sigma_\rho^2(\pi_k) \approx \kappa^2 (1 - \pi_k)^4 \qquad (36)$$

This concludes the derivation of Lemma 1. This result shows that the variance of the importance sampling ratio is monotonically decreasing with the token probability $\pi_k$, as its derivative with respect to $\pi_k$ is negative for $\pi_k \in (0, 1)$.

## B.2 DERIVATION OF PROPOSITION 1

This section provides a detailed derivation for the expected gradient magnitude, $\mathbb{E}[G|\pi]$, as stated in Proposition 1.

### B.2.1 PROBLEM FORMULATION

The policy gradient in a PPO-style objective, ignoring other terms like the KL divergence penalty, is given by:

$$\nabla_\theta J(\theta) = \mathbb{E}_{\pi_{old}} \left[ \nabla_\theta \log \pi_\theta(o_t) \cdot w_t \right] \qquad (37)$$

where $w_t$ is the weighting term after clipping. The policy update for a specific token $k$ with probability $\pi_k$ is proportional to the gradient of the log-probability, $\nabla_\theta \log \pi_\theta(k)$, weighted by $\rho_k \hat{A}_k$, where $\rho_k = \pi_\theta(k)/\pi_{old}(k)$ is the importance sampling ratio and $\hat{A}_k$ is the advantage. The clipping mechanism effectively sets the gradient to zero if the update moves outside the trust region. This can be expressed using an indicator function $\mathbb{I}_{\text{trust}}$:

$$\mathbb{I}_{\text{trust}}(\rho_k, \hat{A}_k) = \begin{cases} 1 & \text{if } (\hat{A}_k > 0 \text{ and } \rho_k \leq 1 + \epsilon_h) \text{ or } (\hat{A}_k < 0 \text{ and } \rho_k \geq 1 - \epsilon_l) \\ 0 & \text{otherwise} \end{cases} \qquad (38)$$

The gradient with respect to the pre-softmax logits for token $k$ is proportional to $(1 - \pi_k)$. We define the gradient magnitude component, $G_k$, for a token $k$ as:

$$G_k := (1 - \pi_k)\rho_k \hat{A}_k \mathbb{I}_{\text{trust}}(\rho_k, \hat{A}_k). \tag{39}$$

To derive its expectation, we make the following simplifying assumptions:

1. The advantage $\hat{A}$ is a random variable following a zero-mean normal distribution: $\hat{A} \sim \mathcal{N}(0, \sigma_A^2)$.

2. For a small update step, the importance sampling ratio $\rho$ is approximately normally distributed around 1: $\rho \sim \mathcal{N}(1, \sigma_\rho^2)$, where $\sigma_\rho^2$ is its variance.

3. The random variables $\hat{A}$ and $\rho$ are treated as independent.

### B.2.2 DECOMPOSING THE EXPECTATION

The expectation of $G_k$ can be split into two parts based on the sign of the advantage:

$$\mathbb{E}[G_k] = \mathbb{E}[(1 - \pi_k)\rho_k \hat{A}_k \mathbb{I}_{\text{trust}}] \tag{40}$$

$$= (1 - \pi_k)\left(\mathbb{E}[\rho_k \hat{A}_k \mathbf{1}_{\hat{A}_k > 0, \rho_k \leq 1 + \epsilon_h}] + \mathbb{E}[\rho_k \hat{A}_k \mathbf{1}_{\hat{A}_k < 0, \rho_k \geq 1 - \epsilon_l}]\right) \tag{41}$$

By the independence of $\rho_k$ and $\hat{A}_k$, we can separate the expectations:

$$\mathbb{E}[G_k] = (1 - \pi_k)\left(\mathbb{E}[\hat{A}_k \mathbf{1}_{\hat{A}_k > 0}]\mathbb{E}[\rho_k \mathbf{1}_{\rho_k \leq 1 + \epsilon_h}] + \mathbb{E}[\hat{A}_k \mathbf{1}_{\hat{A}_k < 0}]\mathbb{E}[\rho_k \mathbf{1}_{\rho_k \geq 1 - \epsilon_l}]\right) \tag{42}$$

### B.2.3 COMPUTING TRUNCATED NORMAL EXPECTATIONS

We use the standard formula for the first moment of a truncated normal distribution. If $Z \sim \mathcal{N}(\mu, \sigma^2)$ and $\alpha = (a - \mu)/\sigma$, then:

$$\mathbb{E}[Z\mathbf{1}_{Z \leq a}] = \mu\Phi(\alpha) - \sigma\phi(\alpha) \tag{43}$$

$$\mathbb{E}[Z\mathbf{1}_{Z \geq a}] = \mu(1 - \Phi(\alpha)) + \sigma\phi(\alpha) \tag{44}$$

where $\Phi(\cdot)$ and $\phi(\cdot)$ are the CDF and PDF of the standard normal distribution, respectively.

**For the Advantage $\hat{A} \sim \mathcal{N}(0, \sigma_A^2)$:** Here, $\mu = 0, \sigma = \sigma_A$.

$$\mathbb{E}[\hat{A}\mathbf{1}_{\hat{A} > 0}] = 0 \cdot (1 - \Phi(0)) + \sigma_A\phi(0) = \sigma_A\frac{1}{\sqrt{2\pi}} \tag{45}$$

$$\mathbb{E}[\hat{A}\mathbf{1}_{\hat{A} < 0}] = 0 \cdot \Phi(0) - \sigma_A\phi(0) = -\sigma_A\frac{1}{\sqrt{2\pi}} \tag{46}$$

**For the IS Ratio $\rho \sim \mathcal{N}(1, \sigma_\rho^2)$:** Here, $\mu = 1, \sigma = \sigma_\rho$. For the upper clip boundary $a_h = 1 + \epsilon_h$, the standardized value is $\alpha_h = \frac{(1 + \epsilon_h) - 1}{\sigma_\rho} = \frac{\epsilon_h}{\sigma_\rho}$.

$$\mathbb{E}[\rho\mathbf{1}_{\rho \leq 1 + \epsilon_h}] = 1 \cdot \Phi(\alpha_h) - \sigma_\rho\phi(\alpha_h) \tag{47}$$

For the lower clip boundary $a_l = 1 - \epsilon_l$, the standardized value is $\alpha_l = \frac{(1 - \epsilon_l) - 1}{\sigma_\rho} = -\frac{\epsilon_l}{\sigma_\rho}$.

$$\mathbb{E}[\rho\mathbf{1}_{\rho \geq 1 - \epsilon_l}] = 1 \cdot (1 - \Phi(\alpha_l)) + \sigma_\rho\phi(\alpha_l) \tag{48}$$

$$= \Phi(-\alpha_l) + \sigma_\rho\phi(-\alpha_l) \quad (\text{since } 1 - \Phi(x) = \Phi(-x) \text{ and } \phi(x) = \phi(-x)) \tag{49}$$

$$= \Phi\left(\frac{\epsilon_l}{\sigma_\rho}\right) + \sigma_\rho\phi\left(\frac{\epsilon_l}{\sigma_\rho}\right) \tag{50}$$

### B.2.4 ASSEMBLING THE FINAL EXPRESSION

Let's substitute these results back into Equation 42. For clarity, let $C_A = \frac{\sigma_A}{\sqrt{2\pi}}$.

$$\mathbb{E}[G_k] = (1 - \pi_k) \left[ C_A \left( \Phi(\tfrac{\epsilon_h}{\sigma_\rho}) - \sigma_\rho \phi(\tfrac{\epsilon_h}{\sigma_\rho}) \right) - C_A \left( \Phi(\tfrac{\epsilon_l}{\sigma_\rho}) + \sigma_\rho \phi(\tfrac{\epsilon_l}{\sigma_\rho}) \right) \right] \tag{51}$$

$$= (1 - \pi_k) \frac{\sigma_A}{\sqrt{2\pi}} \left[ \Phi\left( \frac{\epsilon_h}{\sigma_\rho} \right) - \Phi\left( \frac{\epsilon_l}{\sigma_\rho} \right) - \sigma_\rho \left( \phi\left( \frac{\epsilon_h}{\sigma_\rho} \right) + \phi\left( \frac{\epsilon_l}{\sigma_\rho} \right) \right) \right] \tag{52}$$

Finally, we substitute the expression for $\sigma_\rho$ from Lemma 1, $\sigma_\rho(\pi) = \kappa(1 - \pi)^2$. For a generic token with probability $\pi$, the expected gradient magnitude becomes:

$$\mathbb{E}[G \mid \pi] = (1 - \pi) \frac{\sigma_A}{\sqrt{2\pi}} \left[ \Phi\left( \tfrac{\epsilon_h}{\sigma_\rho(\pi)} \right) - \Phi\left( \tfrac{\epsilon_l}{\sigma_\rho(\pi)} \right) - \sigma_\rho(\pi) \left( \phi\left( \tfrac{\epsilon_h}{\sigma_\rho(\pi)} \right) + \phi\left( \tfrac{\epsilon_l}{\sigma_\rho(\pi)} \right) \right) \right] \tag{53}$$

This matches the form in Proposition 1, where $F(\pi; \kappa, \epsilon_h, \epsilon_l)$ is the term in the brackets. This completes the derivation.

## B.3 DERIVATION OF COROLLARY 1

This appendix provides a detailed asymptotic analysis to prove the gradient dominance reversal phenomenon stated in Corollary 1. We analyze the average gradient expectation magnitude for low- and high-probability tokens and show how their relationship inverts as the degree of off-policy, represented by $\kappa$, changes.

### B.3.1 SETUP AND DEFINITIONS

We partition the token probability space into a low-probability interval $I_L = [0, p_L]$ and a high-probability interval $I_H = [p_H, 1)$, for some thresholds $0 < p_L < p_H < 1$. We then define the average gradient expectation magnitude for each group, assuming a uniform distribution of probabilities within these intervals for analytical tractability:

$$|\bar{G}_L| := \left| \frac{1}{p_L} \int_0^{p_L} \mathbb{E}[G \mid \pi] d\pi \right| \tag{54}$$

$$|\bar{G}_H| := \left| \frac{1}{1 - p_H} \int_{p_H}^1 \mathbb{E}[G \mid \pi] d\pi \right| \tag{55}$$

where $\mathbb{E}[G \mid \pi] = (1 - \pi) \frac{\sigma_A}{\sqrt{2\pi}} F(\pi; \kappa, \epsilon_l)$. Let $C_A = \frac{\sigma_A}{\sqrt{2\pi}}$.

### B.3.2 ASYMPTOTIC ANALYSIS FOR LOW-PROBABILITY TOKENS ($|\bar{G}_L|$)

For low-probability tokens ($\pi \in I_L$), the term $(1 - \pi)^2$ is close to 1. When training becomes sufficiently off-policy, $\kappa$ becomes large, causing $\sigma_\rho(\pi) = \kappa(1 - \pi)^2$ to be large. Consequently, the standardized clipping thresholds $\alpha_h = \epsilon_h / \sigma_\rho(\pi)$ and $\beta_l = \epsilon_l / \sigma_\rho(\pi)$ become small ($\ll 1$). We can approximate the function $F(\cdot)$ using a first-order Taylor expansion for $\Phi(x)$ around $x = 0$, where $\Phi(x) \approx 0.5 + x/\sqrt{2\pi}$, and a zeroth-order expansion for $\phi(x) \approx 1/\sqrt{2\pi}$:

$$F(\pi) = \Phi(\alpha_h) - \Phi(-\beta_l) - \sigma_\rho(\pi)[\phi(\alpha_h) + \phi(-\beta_l)] \tag{56}$$

$$\approx \left( \frac{\alpha_h + \beta_l}{\sqrt{2\pi}} \right) - \sigma_\rho(\pi) \left( \frac{2}{\sqrt{2\pi}} \right) \tag{57}$$

$$= \frac{1}{\sqrt{2\pi}} \left( \frac{\epsilon_h + \epsilon_l}{\sigma_\rho(\pi)} - 2\sigma_\rho(\pi) \right) \tag{58}$$

$$= \frac{1}{\sqrt{2\pi}} \left( \frac{\epsilon_h + \epsilon_l}{\kappa(1 - \pi)^2} - 2\kappa(1 - \pi)^2 \right) \tag{59}$$

Substituting this into the integral for $\bar{G}_L$:

$$\bar{G}_L \approx \frac{C_A}{p_L \sqrt{2\pi}} \int_0^{p_L} (1 - \pi) \left[ \frac{\epsilon_h + \epsilon_l}{\kappa(1 - \pi)^2} - 2\kappa(1 - \pi)^2 \right] d\pi \tag{60}$$

The integral can be solved analytically:

$$\int_0^{p_L} \frac{1}{1-\pi} d\pi = [-\ln(1-\pi)]_0^{p_L} = -\ln(1-p_L) \tag{61}$$

$$\int_0^{p_L} (1-\pi)^3 d\pi = [-\frac{(1-\pi)^4}{4}]_0^{p_L} = \frac{1-(1-p_L)^4}{4} \tag{62}$$

This yields an expression for $\bar{G}_L$ that depends on both $\kappa$ and $1/\kappa$:

$$|\bar{G}_L| \approx \frac{C_A}{p_L\sqrt{2\pi}} \left| \frac{\epsilon_h + \epsilon_l}{\kappa}(-\ln(1-p_L)) - 2\kappa\left(\frac{1-(1-p_L)^4}{4}\right) \right| \tag{63}$$

This shows that for low-probability tokens, the gradient magnitude has competing dependencies on $\kappa$.

### B.3.3 ASYMPTOTIC ANALYSIS FOR HIGH-PROBABILITY TOKENS ($|\bar{G}_H|$)

For high-probability tokens ($\pi \in I_H$), $s = 1 - \pi$ is small. In the near on-policy regime where $\kappa$ is small, $\sigma_\rho(\pi) = \kappa s^2$ is very small. This results in large standardized clipping thresholds $\alpha_h \gg 1$ and $\beta_l \gg 1$. We use the tail approximation for the standard normal CDF, $1 - \Phi(x) \approx \phi(x)/x$ for large $x$.

$$F(\pi) = \Phi(\alpha_h) - (1 - \Phi(\beta_l)) - \sigma_\rho(\pi)[\phi(\alpha_h) + \phi(\beta_l)] \tag{64}$$

$$\approx \left(1 - \frac{\phi(\alpha_h)}{\alpha_h}\right) - \frac{\phi(\beta_l)}{\beta_l} - \sigma_\rho(\pi)[\phi(\alpha_h) + \phi(\beta_l)] \tag{65}$$

Since $\alpha_h, \beta_l \gg 1$, the terms involving $\phi(\cdot)$ decay exponentially fast. The gradient contribution is non-negligible only when clipping is not guaranteed, i.e., when $\sigma_\rho$ is not vanishingly small. A more careful analysis (as sketched in the draft) shows that the integral is dominated by terms that scale inversely with $\kappa$. The integration via a change of variables $u = \epsilon_l/(\kappa s^2)$ and subsequent tail approximation of the resulting integral reveals the dominant behavior:

$$|\bar{G}_H| \approx C_A \frac{c_H}{\kappa} \tag{66}$$

where $c_H$ is a positive constant that aggregates terms related to $p_H, \epsilon_l$, and the value of the asymptotic integral. This key result indicates that the average gradient magnitude from high-probability tokens is largest near the on-policy setting (small $\kappa$) and decays as the policy moves further off-policy.

### B.3.4 THE GRADIENT DOMINANCE REVERSAL

We can now find the critical threshold $\kappa_c$ by equating the magnitudes of the two gradient contributions, $|\bar{G}_L| \approx |\bar{G}_H|$. We are interested in the transition from the regime where 'near on-policy' behavior dominates. For small $\kappa$, the $1/\kappa$ term in $|\bar{G}_L|$ dominates. For large $\kappa$, the $\kappa$ term dominates. The reversal occurs when these competing effects balance out. Equating the dominant behaviors in the respective regimes yields an equation for a crossover point. Let's analyze the equation $|\bar{G}_L| = |\bar{G}_H|$:

$$\frac{C_A}{p_L\sqrt{2\pi}} \left| \frac{L_L}{\kappa} - \kappa Q_L \right| = C_A \frac{c_H}{\kappa} \tag{67}$$

where $L_L = (\epsilon_h + \epsilon_l)(-\ln(1-p_L))$ and $Q_L = (1-(1-p_L)^4)/2$ are positive constants. Multiplying by $\kappa$ and rearranging gives a quadratic equation in $\kappa^2$:

$$\frac{Q_L}{p_L\sqrt{2\pi}}\kappa^2 - \frac{L_L}{p_L\sqrt{2\pi}} = \pm c_H \tag{68}$$

Solving for $\kappa^2$ gives the critical threshold:

$$\kappa_c^2 = \frac{L_L \pm c_H p_L\sqrt{2\pi}}{Q_L} \tag{69}$$

Since $\kappa_c^2$ must be positive, a physically meaningful solution exists. This $\kappa_c$ marks the boundary between two distinct optimization regimes:

- When $\kappa < \kappa_c$ (Near On-Policy): The $1/\kappa$ terms dominate. The analysis shows that the constant factor for $|\bar{G}_L|$ is typically larger than for $|\bar{G}_H|$, resulting in $|\bar{G}_L| > |\bar{G}_H|$. Low-probability tokens provide the dominant gradient signal.
- When $\kappa > \kappa_c$ (Sufficiently Off-Policy): For $|\bar{G}_L|$, the linear term in $\kappa$ would grow, but this is an artifact of an approximation breaking down; in reality, extreme clipping suppresses the contribution, causing $|\bar{G}_L|$ to fall off. Meanwhile, $|\bar{G}_H|$ has already decayed as $1/\kappa$. In this regime, the more stable, albeit decayed, contribution from high-probability tokens becomes dominant, leading to $|\bar{G}_H| > |\bar{G}_L|$.

This completes the theoretical justification for the gradient dominance reversal described in Corollary 1.

## C    Reconciling Heuristics: Probability, Entropy, and Clipping Dynamics

The empirical contradiction is resolved by examining the interplay between probability, entropy, and the clipping mechanism. While a token's probability $\pi$ and the vocabulary's entropy $H$ are correlated, their formal relationship is bounded:

$$\underbrace{-\pi \log \pi - (1 - \pi) \log(1 - \pi)}_{H_{\min}(\pi)} \leq H \leq \underbrace{-\pi \log \pi - (1 - \pi) \log \left( \frac{1 - \pi}{V - 1} \right)}_{H_{\max}(\pi)}$$

Figure 2 right visualizes these bounds and provides the crucial insight: clipped tokens (red crosses) are not randomly distributed but are overwhelmingly concentrated in the low-probability, low-entropy region.

This single observation directly reconciles the conflicting findings:

- Updating **low-probability tokens** (Yang et al., 2025b) is suboptimal because this population is dominated by dynamically unstable, low-entropy tokens. As established in Lemma 1, these tokens are highly susceptible to clipping, which attenuates their gradient contribution.
- Conversely, updating **high-entropy tokens** (Wang et al., 2025b) succeeds because it acts as an effective filter for stability. This heuristic implicitly avoids the heavily clipped region, selecting for tokens from more uniform distributions that are robust to policy updates.

Therefore, entropy is not merely a proxy for low probability but a more discerning indicator of a token's dynamic stability under the clipping pressures that govern off-policy optimization.

## D    Detailed Gradient Analysis

### D.1    Token Property Analysis

To understand the fundamental composition of the policy gradient, we analyze the contributions of different token subsets based on their intrinsic properties. Our analysis, presented in Figure 4 left, isolates the gradient signals originating from distinct populations of tokens, revealing that the update is overwhelmingly dominated by a small, low-probability subset. Consistent with Yang et al. (2025b), tokens with $\pi_k < 0.2$ produce a gradient with a norm 409% that of the full gradient and are exceptionally well-aligned with its direction ($27.8°$ mean principal angle). In stark contrast, high-probability tokens ($\pi_k > 0.8$) contribute a much smaller, less aligned gradient (50% norm, $42.8°$ angle). This confirms that the policy gradient is primarily driven by powerful but potentially unstable signals from the tail of the probability distribution.

The strategy of updating high-entropy tokens, proposed by Wang et al. (2025b), can be understood as an effective heuristic for managing this instability. The high-entropy subset produces a gradient with a substantial magnitude (340% of original) but a weaker directional alignment ($44.2°$) than

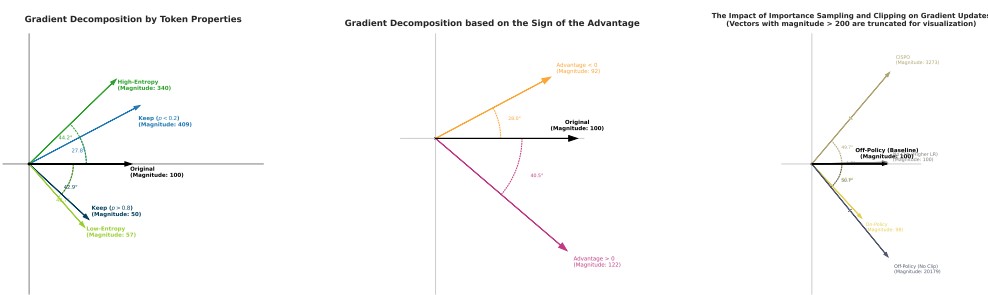

Figure 4: The impact of different key factors on RL updates.

the pure low-probability set. This suggests the high-entropy criterion functions as a filter: it selects for tokens that are informative (often having low to moderate probability) but originate from flatter, more uncertain distributions. As we will show, these flatter distributions are inherently more stable under policy updates. Therefore, the heuristic succeeds by implicitly balancing gradient magnitude against stability. This success underscores the need for a more principled mechanism that can dynamically manage this trade-off.

## D.2 ADVANTAGE SIGN ANALYSIS

The advantage function, $A$, guides learning by signaling whether an action should be reinforced ($A > 0$) or suppressed ($A < 0$). To understand their distinct roles, we analyze the gradient contributions from these two subsets separately, as shown in Figure 4 middle. Our analysis reveals a critical asymmetry: while the gradient norms from positive (122% of original) and negative (92%) advantage samples are comparable, their directional guidance differs markedly. The gradient from negative-advantage samples is substantially better aligned with the total gradient direction (28.0° mean principal angle) than that from positive-advantage samples (40.5°). This indicates that the overall update direction is predominantly dictated by corrective signals from suboptimal actions.

This directional dominance stems from the fundamentally different effects of positive and negative updates. An update with $A > 0$ reinforces a single action, implicitly suppressing all alternatives and potentially narrowing the policy. Conversely, an update with $A < 0$ penalizes a specific action, which effectively redistributes probability mass across the rest of the vocabulary. This latter mechanism, a form of error correction, promotes exploration and enhances policy diversity. The geometric dominance of negative-advantage gradients therefore suggests that learning in complex tasks is driven more by correcting errors than by reinforcing known correct pathways, a process crucial for discovering robust strategies. While Zhu et al. (2025) also noted the importance of negative samples, our work provides a novel perspective by demonstrating their dominant role in shaping the gradient's geometric direction.

## D.3 IMPORTANCE SAMPLING ANALYSIS

To improve sample efficiency, off-policy reinforcement learning corrects for the policy distribution mismatch using Importance Sampling (IS). However, the high variance of the IS ratio necessitates a stabilization mechanism like clipping, which modulates the raw gradient into a stable update signal. To dissect this process, we compare our baseline off-policy GRPO update against several variants (Figure 4 right). The indispensability of clipping is starkly illustrated by its removal: the gradient norm explodes to **20,177%** of the baseline, and its direction severely deviates (**50.7°** mean principal angle). This confirms clipping is crucial not just for controlling magnitude but also for maintaining a stable optimization path.

Perhaps the most revealing finding is the significant directional divergence (**47.4°**) between the on-policy and standard off-policy (GRPO) gradients, despite their nearly identical magnitudes (98% vs. 100%). This highlights a fundamental trade-off: to maintain magnitude stability, GRPO's fixed clip-

ping mechanism systematically filters out certain updates (predominantly from high-variance, low-probability tokens), creating a gradient direction that is geometrically distinct from the on-policy ideal. This implies that while off-policy learning is sample-efficient, its optimization trajectory can substantially differ from its on-policy counterpart.

The specific design of the clipping strategy further modulates this trade-off. For instance, CISPO, which clamps the IS ratio instead of gating the entire gradient to zero, retains more signal from outlier tokens. This results in a substantially larger gradient norm (**3,268%**) and a similarly large directional shift (**49.7°**), reflecting a more aggressive update policy. In contrast, a simple asymmetric clipping scheme ($\epsilon_h > \epsilon_l$), inspired by DAPO, showed negligible impact in our setting ($1.3°$ divergence). These comparisons reveal that the standard clipping in methods like PPO/GRPO is not a neutral stabilizer; it actively shapes the gradient by discarding certain information. This suggests an opportunity for adaptive mechanisms that can better preserve the on-policy direction while retaining off-policy efficiency.

## E  MORE RELATED WORKS

**Reinforcement Learning for Large Language Models.**  Reinforcement learning has evolved from a tool for preference alignment to a key technique for enhancing reasoning capabilities in large language models. Initially pioneered through Reinforcement Learning from Human Feedback (RLHF), RL methods like PPO (Schulman et al., 2017) were used to align models with human preferences using human-annotated data (Ouyang et al., 2022). The landscape shifted dramatically with the emergence of RL with verifiable rewards (RLVR), which leverages objective, automatically verifiable feedback signals instead of subjective human preferences (Lambert et al., 2024). OpenAI's o1 model (Jaech et al., 2024) first showcased that RLVR can effectively incentivize reasoning at scale, particularly in tasks like mathematics and programming. Building on this foundation, researchers developed improved algorithmic methods such as GRPO (Shao et al., 2024) and its variants (e.g., DAPO (Yu et al., 2025)). Subsequent models trained with these methods, including DeepSeek-R1 (Guo et al., 2025a), QwQ (Team, 2025), and AceReason-Nemotron (Chen et al., 2025c), demonstrate that strong reasoning capabilities can emerge through outcome-based optimization with online RL algorithms, establishing RLVR as a promising paradigm for developing reasoning-capable LLMs.

**Token-Level Dynamics and Update Strategies.**  While RLVR has demonstrated strong potential, its effectiveness is often determined by how token-level updates are applied during training. Recent research has explored multiple related directions : entropy- or probability-based methods that incorporate or reduce uncertainty (Wang et al., 2025b; Yang et al., 2025b; Cheng et al., 2025; Cui et al., 2025; Gao et al., 2025; Chen et al., 2025b), modifications to importance-sampling and clipping that stabilize gradients (Roux et al., 2025; Chen et al., 2025a; Su et al., 2025; Zheng et al., 2025a; Wang et al., 2025a), advantage design strategies ranging from negative reinforcement to minimalist or segment-level credit assignment (Zhu et al., 2025; Xiong et al., 2025), and off-policy approaches that reshape update distributions (Yan et al., 2025; Ma et al., 2025; Fu et al., 2025; Arnal et al., 2025). However, studies in this area sometimes yield conflicting conclusions: for example, Wang et al. (2025b) emphasize that high-entropy minority tokens drive effective learning, whereas Yang et al. (2025b) show that low-probability(usually high-entropy) tokens can over-dominate gradients and should be suppressed. Such inconsistencies calls for a deeper, component-wise analysis of the RL update pipeline. While recent work has started to examine how individual training techniques affect RL dynamics (Liu et al., 2025), we still lack a comprehensive framework that can systematically explain how these components work together to drive effective policy updates in LLM reasoning.

## F  TRAINING DETAILS

Our training code is adapted from the VeRL framework (Sheng et al., 2024). To evaluate the effects of on-policy and off-policy settings, we use a training batch size of 512, with mini-batch sizes of 256 and 32 for the respective settings, resulting in 2 update steps for per training batch for near on-policy training, and 16 update steps for off-policy training. The learning rate is set to 1e-6. Additional task-specific hyperparameters can be found in the scripts provided in our GitHub repository.

