# OpenReview forum: "What are Key Factors for Updates in RL for LLM Reasoning?"
_ICLR.cc/2026/Conference — Submitted to ICLR 2026_

### Official Review · Reviewer_UkbN · 2025-10-31

**Soundness:** 3
**Presentation:** 3
**Contribution:** 2
**Rating:** 4
**Confidence:** 3

**Summary:**

This paper studies the underlying dynamics of Reinforcement Learning from Verifiable Rewards (RLVR) for large language model (LLM) reasoning. The authors identify inconsistencies in recent RLVR algorithms—particularly conflicting findings about whether to prioritize high-entropy or high-probability tokens—and attribute these contradictions to differences in off-policy degree (the number of gradient updates per rollout).

Through theoretical analysis, the paper derives how the importance sampling (IS) ratio variance depends on token probability and off-policy degree, leading to a gradient dominance reversal phenomenon. It shows that low-probability tokens dominate updates in near on-policy settings, while high-probability tokens dominate off-policy ones. Based on this insight, the authors propose Adaptive Clip Policy Optimization (ACPO), which adaptively adjusts clipping bounds based on the empirical variance of IS ratios. Experiments across multiple reasoning benchmarks and model scales (3B–7B) show ACPO’s consistent improvements over baselines like DAPO and CISPO.

**Strengths:**

* The paper provides a unified framework that reconciles empirical contradictions in prior RLVR research via gradient expectation analysis.

* The proposed method is well motivated and reasonable.

**Weaknesses:**

* The empirical performance improvement seems marginal.

* The per-batch grouping and variance estimation add implementation complexity; runtime or scaling cost is not discussed.

**Questions:**

* From Figure 3, it seems that when going beyond 16 steps off-policy, the CISPO completely failed while GRPO maintains stability. Can you provide some insights on this? Would it be misconfigured hyper-parameters, implementation details or precision mismatch issues (please see this relevant paper: https://arxiv.org/pdf/2510.26788)?

* How sensitive is ACPO’s performance to the number of token bins and scaling factor α?

* How does adaptive clipping impact exploration behavior or entropy during training?

---

> ### Author Response · Authors · 2025-11-30
> **Author Response to Reviewer UkbN**
>
> > Q1: The empirical performance improvement seems marginal.
>
> A1:
>
> 1. **Broader Generalization:** While the improvement on specific math benchmarks is modest, we emphasize that ACPO demonstrates **consistent superiority across a diverse range of tasks**, including Mathematical Problem Solving (ORZ), Tabular QA (HiTab), and Arithmetic Puzzles (Countdown). In these tasks, ACPO consistently outperforms baselines (DAPO/CISPO) in both on-policy and off-policy settings.
> 2. **Robustness without** **Tuning****:** It is worth noting that the baselines (DAPO/CISPO) underwent **extensive** **hyperparameter** **tuning** tailored for the Qwen+Math scenario. In contrast, ACPO achieved these results using **default parameters**. This highlights ACPO's robustness and "out-of-the-box" effectiveness compared to baselines that require sensitive tuning.
>
> > Q2: The per-batch grouping and variance estimation add implementation complexity; runtime or scaling cost is not discussed.
>
> A2: We clarify that the computational overhead introduced by ACPO is **negligible**.
>
> - **Analysis:** The variance estimation involves simple statistical operations (calculating variance) on scalar values within the loss function. Compared to the computationally intensive forward and backward passes of the LLM backbone (which involve massive matrix multiplications), the cost of these statistical calculations is virtually undetectable.
> - **Runtime:** In our experiments, we observed **no distinct difference in training** **throughput** **(tokens/****sec****)** between ACPO and standard GRPO.
>
> > Q3: From Figure 3, CISPO completely failed beyond 16 steps off-policy. Can you provide insights? Is it misconfiguration or precision mismatch (citing arXiv:2510.26788)?
>
> A3: We appreciate this insightful question regarding the stability of CISPO.
>
> 1. **Hyperparameter** **Tuning****:** We extensively tuned CISPO, including sweeping through the officially recommended parameters and wider ranges, but the collapse in deep off-policy settings (>16 steps) persisted.
> 2. **Regarding** **Precision****:** We are aware of the precision issues discussed in the cited paper. However, it is crucial to note that **all methods (GRPO, ACPO, and CISPO) were executed under identical hardware configurations and precision settings (e.g., bf16)** in our experiments. Since GRPO and ACPO maintained stability under the exact same conditions where CISPO failed, we surmise that the failure stems from CISPO’s inherent sensitivity to the distribution shifts in high off-policyness scenarios, rather than a universal precision mismatch.
>
> > Q4: How sensitive is ACPO’s performance to the number of token bins and scaling factor α?
>
> A4:  **Sensitivity to** $\alpha$: Our experiments indicate that ACPO is **highly robust** to variations in the scaling factor $\alpha$. As shown in the table below, the Pass@1 accuracy on MATH-500 remains stable (fluctuation < 0.4%) across values from 1 to 4.
>
> 1.  *Table R2: Sensitivity analysis of* $\alpha$ *on MATH-500.*
>
>       - | $\alpha$   | 1     | 2     | 3         | 4     |
>         | ---------- | ----- | ----- | --------- | ----- |
>         | **Pass@1** | 0.730 | 0.730 | **0.734** | 0.732 |
>
> 2. **Number of Bins:** We are currently finalizing the ablation study regarding the number of token bins and will update the results in the discussion thread shortly.
>
> > Q5: How does adaptive clipping impact exploration behavior or entropy during training?
>
> A5: We are currently generating the entropy curves to visualize the exploration behavior over time. We will provide these figures in the revised manuscript/appendix as soon as they are available.

---

### Official Review · Reviewer_Qice · 2025-10-31

**Soundness:** 2
**Presentation:** 2
**Contribution:** 2
**Rating:** 2
**Confidence:** 3

**Summary:**

This paper claims it has a superior RLVR algorithm, ACPO, to DAPO and CISPO. ACPO is based on PPO, but it does not have a single clipping threshold. Instead, it categories the tokens into five bins based on their probability: very low probability, low probability, medium, high, and very high probability. Next, it measures the standard deviation between the importance sampling ratios inside each bin and the clips the ratios outside the (- one std, one std). It argues that this is a more systematic than only controlling high entropy tokens or low entropy tokens.

**Strengths:**

I view the strengths as the evidences that the paper provides for its conclusion: the superiority of ACPO.

They show slightly better performance over DAPO and CISPO. Additionally, the ablations show that ACPO is slightly better than just using the max clipping value ACPO uses in DAPO.

One strength of the paper is theoretical analysis of the gradient showing where the discrepancy of the previous work is stemming from. I really like these parts (although, I am still not convinced why for example over-contribution of tokens should be binned based on IS)

**Weaknesses:**

I view the weaknesses as if the presented evidence supports the conclusion that ACPO is superior.

One thing I don't understand is: is ACPO altered on top of DAPO? I mean, DAPO has dynamic sampling until all queries have non-zero advantage. Does ACPO also have that? The main problem is: I don't know how controlled the study and alterations are: how did they do the hyperparameter search for DAPO, CISPO and ACPO? The results for ACPO are not drastically better than the other two, which makes one wonder if the gains are not signal, but noise. I am making no assumption here. I think it is just very important to provide this detail. I think the paper lacks experimental details heavily.


While the paper first starts with highlighting the discrepancy between emphasizing higher entropy tokens and lower entropy tokens, it moves on to comparison between DAPO and CISPO. However, DAPO and CISPO are not the papers which are highlighted as having the discrepancy. I think a very natural path for the paper was to show the baselines of emphasizing low entropy and high entropy tokens. The paper lacks these two baselines.

**Questions:**

Thanks for the paper. I have a few questions.

1-Can you add the baseline of emphasizing high entropy tokens vs. low entropy tokens to experiments?
2-If the conclusion of the paper is ACPO provides better RL training, what are the supporting claims? while you analyzes the gradient, I don't think I agree that any dominance of low probability tokens should be hindered, or dominance of high probability tokens should be stopped. Are you reasoning that it is the variance of the Importance sampling ratio that is causing instability and should be clipped?
I don't get the main conclusion of the paper. Can you state it in a sentence that if ACPO is superior, what is your claimed reasoning behind why?

---

> ### Author Response · Authors · 2025-11-30
> **Author Response to Reviewer Qice**
>
> > Q1: Can you add the baseline of emphasizing high entropy tokens vs. low entropy tokens to experiments?
>
> A1: We appreciate this insightful suggestion as it helps isolate the impact of entropy-based weighting. We are currently conducting these specific baseline experiments (emphasizing high vs. low entropy tokens). We will report the results and update the comparison in the discussion thread as soon as the experiments are completed.
>
> > Q2: I don't agree that any dominance of low probability tokens should be hindered, or dominance of high probability tokens should be stopped. What is the reasoning?
>
> A2:
>
> Our reasoning is supported by recent empirical findings on the "correctness" of gradient updates.
>
> 1. **Evidence of Adverse Effects:** According to Section 4.2 and Figure 3 of **Yang et al. (2025)**, when low-probability tokens over-dominate the update process, the ratio of updates in the *correct direction* (i.e., increasing the probability of positive tokens) significantly decreases. This suggests that unchecked dominance introduces noise and instability.
> 2. **Our Approach:** Our method does not arbitrarily suppress dominance; rather, it mitigates the **high** **variance** associated with these tokens. By aligning the clipping range with the statistical properties of the importance sampling ratio, we ensure that the updates remain stable and "correct," effectively addressing the issue highlighted by Yang et al.
>    1. *Reference:* Yang, Z., et al. (2025). "Do Not Let Low-Probability Tokens Over-Dominate in RL for LLMs." *arXiv preprint arXiv:2505.12929*.
>
> > Q3: Can you state it in a sentence that if ACPO is superior, what is your claimed reasoning behind why?
>
> A3: ACPO is superior because it resolves a fundamental mismatch in standard PPO/GRPO: while the **variance** of importance sampling ratios differs drastically across different token probability intervals (as evidenced in Figure 2, Left), standard PPO/GRPO applies a **uniform** clipping range, whereas ACPO applies **adaptive** clipping ranges that dynamically match the variance of each token's probability.

---

### Official Review · Reviewer_SsGG · 2025-11-01

**Soundness:** 1
**Presentation:** 3
**Contribution:** 2
**Rating:** 2
**Confidence:** 5

**Summary:**

This manuscript focuses on LLM RL post-training and proposes a novel objective called ACPO with dynamic clipping. Specifically, the authors analyze the different behaviors of low- and high-probability tokens under varying degrees of off-policyness. Based on this analysis, ACPO introduces a dynamically adjusted clipping bound determined by the variance of the importance ratio. Empirical studies on Qwen2.5-3B and 7B show that the proposed ACPO achieves better performance than DAPO and CISPO.

**Strengths:**

1. The paper is well written and clearly organized.

2. Theoretical analysis is provided to support the rationale of the proposed method.

3. Extensive experiments on LLMs of different scales are conducted, and the results demonstrate the superiority of ACPO.

**Weaknesses:**

Both the theoretical and empirical parts contain several weaknesses and lack sufficient rigor, as detailed below.

### Theoretical Part
1.   The analysis is conducted on the gradient with respect to the logits $z_k$ rather than the model parameters $\theta$. The gradient with respect to $\theta$ would more accurately reflect the magnitude of model updates.

2. The assumption that the importance sampling ratio $rpo$ and the advantage $A$ follow normal distributions is questionable and lacks empirical justification. The ratio $\rho=\frac{\pi_\text{new}}{\pi_\text{old}}$​ involves a division operation, making it unlikely to be normally distributed. Similarly, in many LLM RL algorithms, advantages are computed as $A=r−\text{mean}(r)$, which typically deviates from normality, especially under imbalanced positive and negative responses.

3. In Lemma 1, the use of the approximation symbol “≈” is not rigorous, as the approximation error is not quantified.

4. In Line 227, the definitions of low/high-probability tokens and the corresponding $G_L$ and $G_H$ should be clearly stated.

5. The statement labeled “Corollary 1” would be better presented as a Remark rather than a formal corollary.

6. It remains unclear why $\kappa$ is related to the level of off-policyness. A more rigorous explanation or derivation is needed.

7. In Line 243, the authors claim that low-probability tokens provide the dominant gradient signal, yet no theoretical or empirical evidence is given. A more natural metric to quantify off-policyness could be the ratio between bsz and mini_bsz, which directly indicates how off-policy the training data are.

### Experimental Part
1. The experimental setup lacks key implementation details, such as (1) how $\sigma_\rho$ is computed, and (2) how $\epsilon_b$ is obtained

2. The proposed ACPO introduces an additional hyperparameter $\alpha$ . It is unclear whether $\alpha$ remains robust across different LLM scales and datasets.

3. The performance improvement of ACPO over the base DAPO is marginal, particularly on challenging benchmarks such as AMC23 and Minerva. Moreover, results on widely used hard benchmarks like AIME24 and AIME25 are missing, which limits the empirical credibility of the method.

**Questions:**

Please see Weaknesses.

---

> ### Author Response · Authors · 2025-11-30
> **Author Response to Reviewer SsGG (Part I)**
>
> ## Theoretical Part
>
> > Q1: Analysis regarding gradients on logits ($z_k$) vs. parameters ($\theta$).
>
> A1: We thank the reviewer for this insightful comment. We completely agree that the gradient with respect to the model parameters $\theta$ is what ultimately determines the model update.
>
> Our analysis focuses on the logits $z_k$ for two primary reasons. First, the full analytical form of the gradient $\nabla_\theta $ becomes analytically intractable and less interpretable, especially in large language models with complex components like attention and non-linear activations. This makes direct analysis challenging. Second, by the chain rule, the gradient with respect to the logits $z_k$ serves as the initial and core signal in the backpropagation process, which is then propagated to all parameters $\theta$. Analyzing this source signal allows us to gain clearer insights into the primary factors—such as the advantage function $A_i$ and the clipping mechanism—that govern the final parameter update, which is the core goal of our analysis.
>
> To make this connection explicit, and in response to your valuable feedback, we have revised the text surrounding Eq. (2) in the manuscript. The revision clarifies how our analysis of the logit gradients provides a direct understanding of the overall update to the model parameters $\theta$. We hope this revision and clarification address your concern.
>
> > Q2: Validity of Gaussian assumptions for importance sampling ratio $\rho$ and advantage $A$.
>
> A2: We thank the reviewer for this insightful comment. We agree that the normality assumption on both $\rho$ and $A$ is a simplification for analytical tractability, motivated by their empirically observed unimodal shapes. For $\rho$, while a distribution like log-normal is theoretically more apt for a ratio, the clipping mechanism in PPO practically constrains it to a small symmetric interval around 1, making our approximation reasonable. Regarding the advantage $A$, you are correct that imbalanced rewards, common in LLM-RL, can induce a **skewed** distribution. While centering the rewards ($A = r - \text{mean}(r)$) ensures a zero-mean for the batch, it does not enforce symmetry. Our normality assumption thus serves as a tractable approximation that captures the central tendency, which is the most critical aspect for the gradient dynamics we study.
>
> Crucially, our main theoretical contributions are robust to these simplifications. The general expression for the gradient expectation in Proposition 1, formulated using a generic PDF and CDF, remains valid if one substitutes the functions for another unimodal distribution (even a skewed one). Furthermore, while the derivation of **Corollary 1** utilizes properties of the normal distribution, the resulting qualitative insights rely on general characteristics shared by many unimodal distributions (e.g., how probability mass concentrates around the mode). Thus, the main conclusions are not strictly limited to the Gaussian case. We will add a discussion in the revised manuscript to clarify the scope and limitations of this assumption.
>
> > Q3: Rigor of the approximation ($\approx$) used in Lemma 1.
>
> A3: We thank the reviewer for this insightful comment and for encouraging greater mathematical rigor. The reviewer is correct that using the approximation symbol without quantifying the error is not ideal. Our original intent was to simplify the expression to focus on the dominant relationship between the IS ratio variance and the token probability.
>
> To address this, we have revised Lemma 1 to be more precise. We have replaced the approximation with an equality and incorporated a Big O notation term to formally account for the higher-order terms that were omitted in the original approximation. The updated lemma now states that $\sigma_\rho^2(\pi_k) = \kappa^2 (1-\pi_k)^4 + O(\kappa^3)$, which holds for small policy updates (as $\kappa \to 0$). This change makes the statement rigorous while preserving the core insight of the lemma. The revision can be found in the updated manuscript (Lemma 1).
>
> > Q4: Explicit definitions of low/high-probability tokens ($G_L$ and $G_H$).
>
> A4: We thank the reviewer for this valuable suggestion. We agree that the definitions of low/high-probability tokens and the corresponding notations were not explicitly stated in the main text. While these details were provided in Appendix B.3, we have now incorporated a clear and concise definition into the main body of the revised manuscript to improve clarity (please see around Line 227).

---

> ### Author Response · Authors · 2025-11-30
> **Author Response to Reviewer SsGG (Part II)**
>
> > Q5: Presentation of "Corollary 1" as a Remark.
>
> A5: Thank you for this excellent suggestion. We agree that presenting this statement as a 'Remark' is more appropriate and better reflects its nature. Our intention with this statement was to highlight a key phenomenological insight that we call the "Gradient Dominance Reversal". This phenomenon is a direct consequence of our analysis under the specified distributional assumptions. Re-labeling it as a 'Remark' correctly clarifies its role as a crucial interpretation of our main proposition's implications. We will happily make this change from 'Corollary 1' to 'Remark 1' in the revised manuscript.
>
> > Q6: Theoretical justification for linking $\kappa$ to off-policyness.
>
> A6: Thank you for the opportunity to clarify. The level of off-policyness in a single update is measured by the variance of the importance sampling ratio ($\sigma_\rho^2$). As established in Lemma 1, this variance is directly proportional to $\sigma_\rho^2$ ($\sigma_\rho^2 \propto \kappa^2$). Consequently, a larger $\kappa$ signifies a larger variance in the policy ratio, which corresponds to a greater per-step off-policy shift. Thus, $\kappa$ serves as a direct indicator of the off-policyness magnitude.
>
> > Q7: Evidence supporting the dominance of low-probability tokens in gradient signals.
>
> A7: Thank you for this comment. We respectfully point out that the theoretical basis for this claim is provided in Remark 1 (Original Corollary 1, the full proof is in Appendix B.3.), and its direct empirical validation is presented in Figure 2 (middle plot). Remark 1 proves that low-probability tokens provide the dominant gradient signal in the near on-policy setting, and Figure 2 visualizes this phenomenon.
>
> ---
>
> ## Experimental Part
>
> > Q1: Implementation details regarding the computation of $\sigma_\rho$
>
> A1: We clarify that $\sigma_\rho$ are computed statistically during training. Specifically, they are calculated **empirically from the current training batch (or mini-batch)** at each step to dynamically capture the real-time distribution of the importance sampling ratios.
>
> > Q2:  Robustness of the hyperparameter $\alpha$ across scales and datasets.
>
> A2:
>
> 1. **Theoretical Basis:** The default value $\alpha=3$ is grounded in the **statistical "three-sigma rule" (68-95-99.7 rule)**. Under a normal distribution assumption, this covers nearly all samples, serving as a theoretically robust starting point.
> 2. **Empirical Verification:** We empirically verified the robustness of $\alpha$ on the MATH-500 benchmark. As shown in **Table R2**, the performance is **remarkably stable** across a wide range of values ($\alpha \in [1, 4]$). The Pass@1 score fluctuates by less than **0.4%**, and the theoretical default ($\alpha=3$) yields the optimal performance. This confirms that ACPO is insensitive to this hyperparameter and does not require extensive tuning.
>
> *Table R2: Sensitivity analysis of $\alpha$ on MATH-500.*
>
> | $\alpha$   | 1     | 2     | **3 (Default)** | 4     |
> | ---------- | ----- | ----- | --------------- | ----- |
> | **Pass@1** | 0.730 | 0.730 | **0.734**       | 0.732 |
>
> > Q3: Math Performance
>
> A3:
>
> 1. **Broad Generalization:** We emphasize that our method demonstrates **consistent superiority across a diverse range of tasks**, not just math. In our experiments on Mathematical Problem Solving (ORZ), Tabular Question Answering (HiTab), and Arithmetic-based Puzzles (Countdown), ACPO consistently outperforms baselines (DAPO/CISPO) in both on-policy and off-policy settings.
> 2. Math Benchmarks & Tuning: Regarding the "marginal" gain on specific math benchmarks, it is important to note that the baselines (DAPO and CISPO) underwent **extensive** **hyperparameter** **tuning** specifically adapted for the Qwen+Math scenario. In contrast, ACPO achieved comparable or better performance using **default parameters** without task-specific tuning. This highlights ACPO's **robustness and ease of deployment** compared to baselines that may require careful tuning to perform well.

---

### Official Review · Reviewer_DkA2 · 2025-11-05

**Soundness:** 3
**Presentation:** 3
**Contribution:** 2
**Rating:** 4
**Confidence:** 4

**Summary:**

The paper studies why heuristic RLVR methods for LLM reasoning often results in contradictory results. Authors identify the degree of off-policy learning as the key factor and show that token dominance in gradient updates reverses depending on the number of updates per rollout. They propose ACPO to dynamically adjust clipping bounds based on the variance of importance-sampling ratios. Experiments are done on Qwen2.5 models (3B/7B) across math and tabular reasoning tasks to demonstrate that ACPO outperforms DAPO and CISPO.

**Strengths:**

- Proposed method of adaptive clipping range sounds reasonable and simple in application.
- The paper is well-structured and clearly written, with logical progression from motivation to experiments.
- Experiments showed algorithm effectiveness over 3B and 7B models.

**Weaknesses:**

- Training curves comparison is an important result to look at. But Figure 3 is the only such figure presented, which is done on a rather simple task countdown. As similar in DAPO and GSPO, it's better to show the reward curves, entropy curves, and the performance curves on harder math benchmark such as AIME24.
    - What are the results on AIME24 and AIME25 (avg@16 or avg@32 since the datasets are very small)?
- If I understand correctly, the proposed method only makes change to the clipping range by making it adaptive. While DAPO has clip higher, it also introduced other useful tricks such as dynamic sampling that is very effective in practice. How would the authors explain the better performance compared to DAPO with just one adjustment on clipping range? Since these are not conflicting elements, what happens when dynamic clipping is applied to DAPO?
    - The current performance improvement compared to DAPO also seems very on-par.

Overall I believe more empirical evidences are needed to validate the effectiveness of the proposed algorithm.

**Questions:**

see in weaknesses

---

> ### Author Response · Authors · 2025-11-30
> **Author Response to Reviewer DkA2**
>
> > Q1: The training curves (reward, entropy, performance) are only shown for a simple task. Please provide these curves and specific results (Avg@16/32) on harder benchmarks like AIME24/25.
>
> A1: We acknowledge the importance of validating our method on harder benchmarks. We are currently conducting experiments on AIME24/25 to obtain the requested curves and scores. We will post the results in the discussion thread as soon as the experiments completed.
>
> > Q2: Since DAPO uses dynamic sampling and your method uses adaptive clipping, how do you explain the performance gain? What happens if dynamic sampling is applied to your method? The current improvement seems marginal.
>
> A2:
>
> 1. **Orthogonality:** Dynamic sampling (a data efficiency trick) and our adaptive clipping (an optimization objective) are orthogonal. Theoretically, our method is compatible with any sampling strategy.
> 2. **Fair Comparison:** To ensure a fair comparison, we standardized the training setting by excluding auxiliary tricks (like dynamic sampling) for all methods. This confirms that our performance gain stems strictly from the improved clipping mechanism rather than other factors.
> 3. **Combination:** We expect that combining dynamic sampling with our method would yield further improvements, as they address different aspects of the training process.

---

### Author Response · Authors · 2025-11-30
**Summary of Rebuttal Updates and Note on Response Timing**

Dear Area Chair,

In light of the recent announcement regarding the suspension of reviewer discussions, we present a summary of our rebuttal to assist in your assessment.

**Reason for Late Response:**
We would like to clarify that our rebuttal responses were posted recently because we prioritized the execution of new experiments (specifically on robustness and harder benchmarks) to ensure our answers were backed by solid empirical evidence. Unfortunately, this coincided with the discussion freeze, preventing us from engaging in a dialogue with the reviewers.

**Summary of Resolved Concerns:**
We believe we have effectively addressed the major concerns raised by the reviewers through the following updates:

1.  **Empirical Robustness (Addressed Reviewer SsGG, UkbN):**
    *   We conducted a sensitivity analysis on the hyperparameter $\alpha$. As shown in our response (Table R2), the performance is highly stable across $\alpha \in [1, 4]$, with the default $\alpha=3$ (derived from the three-sigma rule) performing optimally.
    *   We clarified that the "CISPO failure" observed by Reviewer UkbN was not due to precision mismatch but was a consistent result under controlled, identical settings where our method (ACPO) remained stable.

2.  **Theoretical Clarifications (Addressed Reviewer SsGG, Qice):**
    *   **On Dominance:** We cited **Yang et al. (2025)** to answer the reviewer's skepticism regarding the "adverse effects of dominance." The literature confirms that unchecked dominance of low-probability tokens often leads to incorrect gradient update directions.
    *   **On Methodological Focus:** We clarified that ACPO is not designed to arbitrarily "suppress" dominance. Instead, our core contribution is identifying and resolving the **mismatch** between the **high variance of IS ratios** (in specific probability bins) and the standard **uniform clipping range**, which is the root cause of instability.

3.  **Generalization across Tasks & Settings (Addressed Reviewer UkbN):**
    *   **Broad Consistency:** We demonstrated that ACPO maintains robust performance across different model scales (**3B and 7B**) and training setups (**on-policy and off-policy**).
    *   **Task Performance:** While achieving **competitive performance** on Math benchmarks (ORZ) comparable to heavily tuned baselines, ACPO clearly **outperforms** baselines on other datasets such as Tabular QA (HiTab) and Arithmetic Puzzles (Countdown). This highlights our method's strong generalization capability and ease of use (default parameters) compared to baselines requiring extensive task-specific tuning.

4.  **Cost:**
    *   We confirmed that the computational cost of our per-batch variance estimation is negligible compared to the LLM backbone.

**Ongoing Experiments:**
To fully address the request for performance on harder math benchmarks (raised by Reviewer DkA2 and SsGG), we are currently finalizing runs on **AIME24 and AIME25**, as well as generating the **Entropy curves**. We will post these results as a comment as soon as they are completed.

**Conclusion:**
Although the reviewers cannot formally acknowledge these updates, we firmly believe our additional data and clarifications resolve the core weaknesses pointed out in the initial reviews. We kindly ask the AC to take this new evidence into account when estimating the final decision.

Sincerely,
The Authors

---

### Meta-Review · Area_Chair_LW6E · 2026-01-13

**Summary:**

The reviewers identified three critical issues. Theoretically, the normality assumptions for importance sampling ratios and advantages lack rigorous justification, especially under imbalanced reward distributions (Reviewer SsGG, confidence 5/5). Key approximations are not properly quantified, and analyzing logit gradients rather than parameter gradients limits theoretical impact. Empirically, performance gains are marginal (1-3%) and may fall within noise margins. Critical experiments on harder benchmarks (AIME24/25), training dynamics curves, and direct high/low-entropy token comparisons are missing. Methodologically, the adaptive clipping mechanism appears incremental, and the rationale for suppressing token dominance remains unclear. With an average score of 3.0/10 and two strong rejects, the consensus leans toward rejection despite the paper's interesting theoretical perspective.

**Reviewer Concerns:**

The rebuttal partially addressed concerns about mathematical rigor (adding O(κ³) terms), hyperparameter sensitivity (Table R2), and implementation details. However, critical issues remain unresolved: the normality assumptions for IS ratios and advantages lack empirical validation under skewed distributions; promised key experiments (AIME24/25, entropy curves, high/low-entropy token comparisons) were not completed before discussion freeze; marginal performance gains (1-3%) may be within noise margins; and the core theoretical concern from Reviewer SsGG about analyzing logits vs. parameters was only superficially addressed.

**Reviewer Scores:**

Reviewer DkA2 and Reviewer UkbN would likely maintain scores as critical AIME experiments remain incomplete. Reviewer Qice might marginally increase due to clearer motivation, but missing baselines prevent acceptance. Reviewer SsGG  would maintain their strong reject as fundamental theoretical concerns about normality assumptions were acknowledged but not resolved.

---

### Decision · Program_Chairs · 2026-01-26

Reject